# Machine learning in Alzheimer's disease genetics

Traditional statistical approaches have advanced our understanding of the genetics of complex diseases, yet are limited to linear additive models. Here we applied machine learning (ML) to genome-wide data from 41,686 individuals in the largest European consortium on Alzheimer's disease (AD) to investigate the effectiveness of various ML algorithms in replicating known findings, discovering novel loci, and predicting individuals at risk. We utilised Gradient Boosting Machines (GBMs), biological pathway-informed Neural Networks (NNs), and Model-based Multifactor Dimensionality Reduction (MB-MDR) models. ML approaches successfully captured all genome-wide significant genetic variants identified in the training set and 22% of associations from larger meta-analyses. They highlight 6 novel loci which replicate in an external dataset, including variants which map to *ARHGAP25*, *LY6H*, *COG7*, *SOD1* and *ZNF597*. They further identify novel association in *AP4E1*, refining the genetic landscape of the known *SPPL2A* locus. Our results demonstrate that machine learning methods can achieve predictive performance comparable to classical approaches in genetic epidemiology and have the potential to uncover novel loci that remain undetected by traditional GWAS. These insights provide a complementary avenue for advancing the understanding of AD genetics.

Genome-wide association studies (GWAS) have enabled huge progress in identifying variants associated with the risk of developing Alzheimer's disease (AD)[1]. Polygenic risk scores (PRS) based on these variants have greatly improved prediction of disease status[2]. However, inherent to GWAS and PRS are the assumptions that variants are independent predictors, linearly associated with the outcome, and therefore combine additively within and between loci[3], with no interactions occurring between variants, or between genes and other risk factors. While such simplifying genetic assumptions have proved fruitful across a range of diseases and disorders[4,5], they are at odds with biological evidence in AD that disease heterogeneity and responses from cells such as microglia are dependent on *APOE* status[6–9]. Further, there is genetic evidence suggesting that different variants are associated with the disease depending on *APOE* status[10–13] and age at diagnosis or assessment[14–16]. As GWAS sample size increases and PRS approach limits on predictive performance, alternative modelling approaches

are essential to maximise discoveries from existing data and enable a deeper understanding of AD genetics.

The confluence of increasingly large genetic data[17], readily available computational resources, and mature methodologies presents a key opportunity for addressing this at scale by applying flexible data-driven machine learning (ML) models. Several studies have applied ML to the genetics of brain disorders and have been recently summarised[18,19]. Previous ML attempts have been impacted by high risk of bias[20] and population stratification[21], while AD studies in particular have been hampered by low sample size[18], leaving a gap for comprehensive, large-scale studies that rigorously apply ML to genome-wide data. We conducted the largest genome-wide ML study in AD to date, marking a pivotal moment in the field. Our study presents a reproducible, bias-aware approach for ML model development, validation, and confounder adjustment. We trained three of the most prominent approaches in the field to compare predictive accuracy and uncover

✉e-mail: kristel.vansteen@uliege.be; cornelia.vanduijn@ndph.ox.ac.uk; EscottPriceV@cardiff.ac.uk

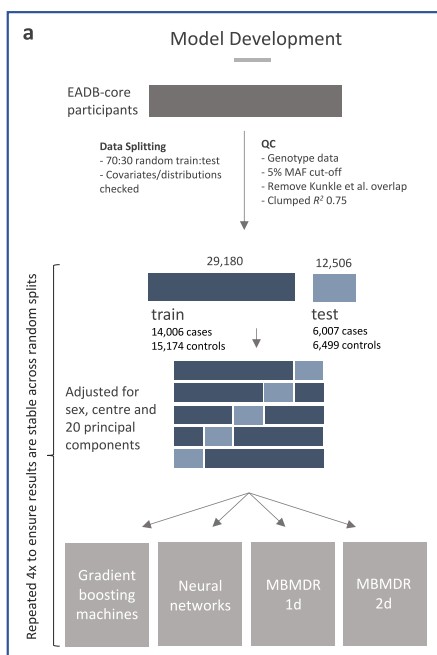

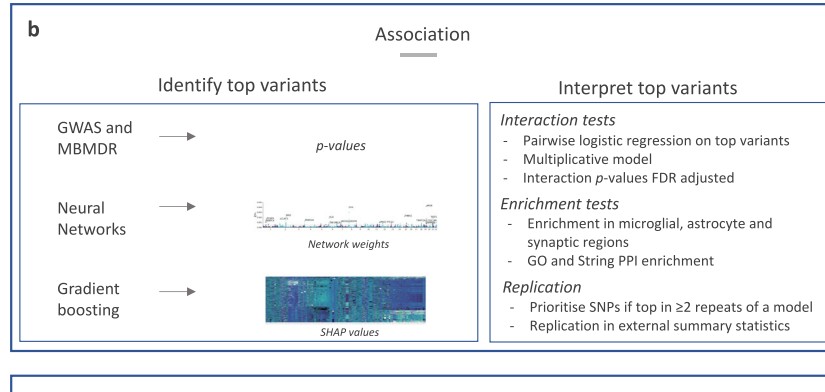

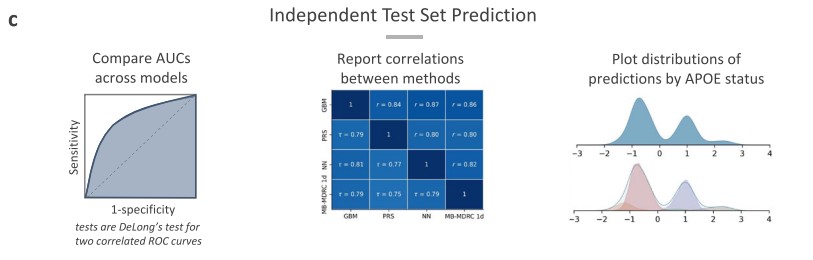

**Fig. 1 | Methods overview.** Data was separated into an initial balanced random split before model selection (cross-validation and hyperparameter tuning) in the training split (**a**). All models were subsequently evaluated for association (annotation, enrichment analysis, interaction testing and replication; **b**) and prediction (AUC and correlations; **c**). Interaction tests report p-values from the Wald test in logistic regression (two-sided) as standard, after correction for multiple testing The full pipeline was run four times per model to assess robustness. For prediction, AUC values, statistical tests, and correlation analyses are based on the initial train-test split. For association, variants were prioritised if they appeared in the top SNP selection in at least two repeats.

novel AD-associated risk loci which were not identified by traditional GWAS.

## Results

### Prediction

Gradient boosting, neural networks, MB-MDRC 1 d, and PRS models were compared for discovery and prediction (Fig. 1). Prediction of AD status between models was highly correlated in the test set, having pairwise correlations between $r = 0.80$ and $r = 0.87$ (Fig. 2). The highest correlations were observed for GBM-NN ($r = 0.87$) and GBM-MB-MDRC 1 d ($r = 0.86$). The weakest correlations were between NNs and PRS ($r = 0.80$). PRS was most strongly correlated with GBMs ($r = 0.84$). For discrimination between cases and controls, the highest AUC of 0.692 (95% CI: 0.683-0.701) was obtained with gradient boosting, and was not significantly different to an AUC of 0.689 (0.679-0.698) for PRS (Fig. 2, Supplementary Data 4). The AUCs remained within the 95% CI when we excluded any imputed variants from the data, indicating no risk of bias from their inclusion (GBMs: 0.683, NN: 0.674, and MBMDRC-1d: 0.667 without imputed variants). Predictions remained stable across repeats with different random train-test splits (Supplementary Data 4) and across the cohorts in the data (Supplementary Fig. 6). All models have a greater proportion of females in those predicted to be a case, reflecting the underling sex differences in the data (59% female, 62% female in cases, 57% in controls), except for GBMs which have a similar proportion in both cases and controls (Fig. 2i).

### Identification of AD-associated loci

SNPs prioritized by ML approaches were required to appear in at least two random train-test splits, ensuring more robust associations; they include both known (Table 1) and novel variants (Table 2). Gradient boosting machines correctly distinguished the *APOE* haplotypes (Fig. 3a, showing distinct clusters). Figure 3b further highlights modelling of the *APOE* region across methods, wherein GBMs and NNs correctly identify causal SNPs. Known loci including *CR1, BIN1, IDUA, OTULIN, RASA1, RASGEF1C, CLU, ABCA1, MS4A\*, PICALM, ABCA7, APOE,*

*and CASS4* (Table 1) are highlighted by ML models (Fig. 3c). In addition, several novel loci were identified with putative biological evidence for association with AD (*ARHGAP25, COG7, LINC00924/LOC105369212, LY6H, SOD1* and *ZNF597*) which were replicated in Jansen et al.[22] (Table 2). Association of an exonic missense variant was also highlighted in *AP4E1*, within 500 kb of the known *SPPL2A* locus (Table 2). Neural networks (NN) detected known loci in *APOE, BIN1, CYP27C1, ABCA1* and *ABCA7* (Fig. 3c) and an additional novel locus (*SOD1*), which also replicated in Jansen et al.[22] (Table 2). MB-MDR 1 d identified SNPs in 24 genes, the majority of which map to the *APOE* region, for at least two train-test splits. Of these, 20 were identified by every possible split, indicating highly stable results. MB-MDR 1 d identified SNP-SNP pairs in the *APOE* region gene consistently through every train-test split. Single train-test splits also find genes outside of Chromosome 19 (Supplementary Fig. 7). The majority of candidate novel loci were identified by GBMs, which find multiple loci with evidence for association with AD-related traits such as cognition, pTau, AD age-at-onset and neurofibrillary tangles from previous GWAS (see Table 2 and Supplementary Data 5).

Since ML may identify non-linear SNP-SNP interactions, pairwise interaction tests were performed for all top SNPs identified using machine learning models (Tables 1 and 2). Of 17,205 SNP-SNP pairwise interactions, 13 pairs were significant under a standard regression framework (encoded as a multiplicative interaction) after accounting for multiple testing and excluding pairs where both are within the *APOE* region. The two SNP-SNP pairs with the strongest evidence for association were between SNPs rs405509 and rs600550 (beta = 0.058, $p_{FDR} = 6.8 \times 10^{-5}$), and SNPs rs405509 and rs12421663 (beta = −0.056, $p_{FDR} = 1.8 \times 10^{-4}$), both of which involve SNPs in the *APOE* and *MS4A\** regions (Supplementary Data 6, Supplementary section 2.5).

### Overlap of genes associated with disease risk across methodologies

Known lead variants in *APOE* and *BIN1* were the most important predictors in GBMs and NNs, with the SNPs used to derive *APOE*

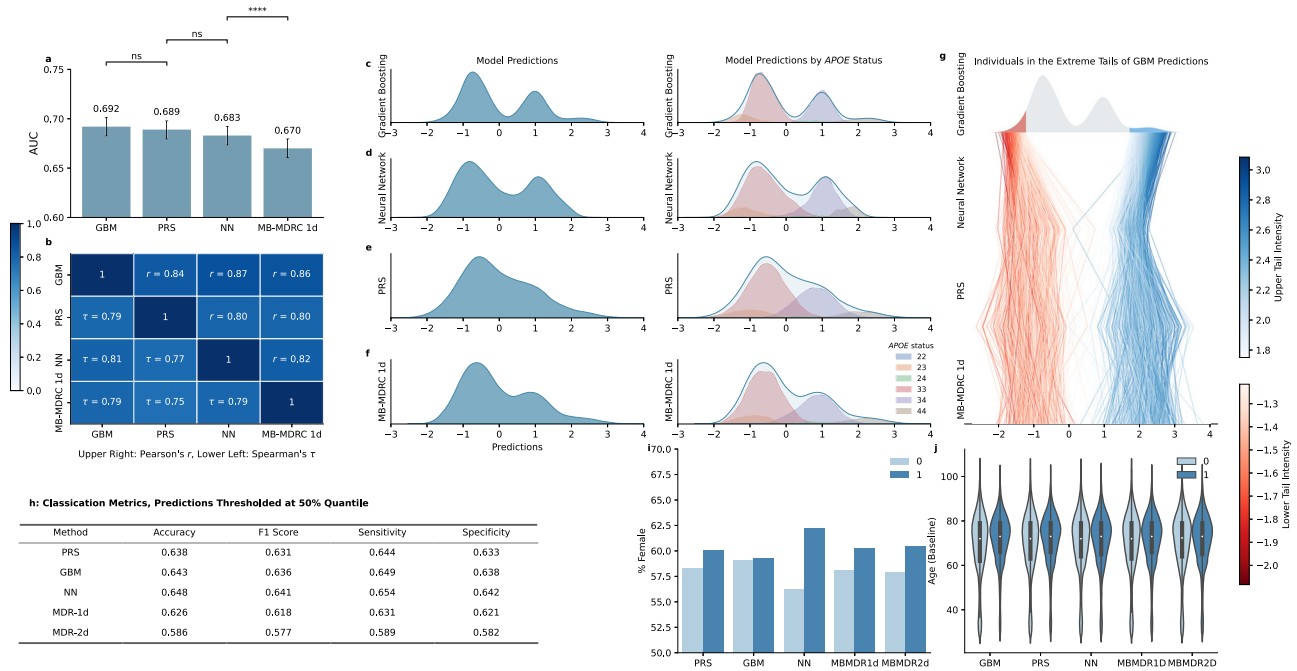

**Fig. 2 | Prediction from ML models in the test split for the most predictive models trained with the *APOE* region included.** The top two most predictive approaches (GBM, PRS) were not significantly different by AUC, as measured by DeLong's test, though prediction from MB-MDRC 1 d was significantly below other methods (**a**). Bars show the AUC from a single test split for each model, where whiskers are 95% CIs from the pROC package. Unadjusted *p*-values from tests (ns: not significant, ****: *p* < 0.0005) are annotated on panel a for DeLong's two-sided test for correlated ROC curve; see supplementary Data 3 for exact values. Model predictions showed strong correlation, though correlation of the ranks is lower (**b**). Distributions of covariate-adjusted predictions for the most predictive approaches

are similar but show a more distinct multimodal distribution for GBMs (**c**), NNs (**d**) and MB-MDRC 1 d (**f**) compared to PRS (**e**), illustrating stronger influence of *APOE* risk alleles on predictions. Panel (**g**) shows the consistency across methods for individuals' prediction scores, where the participants in the 5% extreme tails of GBM predictions are followed across predictions from NN, PRS and MB-MDRC 1 d. Classifications metrics are given in (**h**). Model predictions broken down by covariates are shown in (**i**) and (**j**), where dark blue indicates predicted cases, and light blue predicted controls. Box plots in (**j**) show the median (center line), the 25th and 75th percentiles (box limits), and the whiskers which extend to 1.5 times the interquartile range.

## Table 1 | 18 known loci prioritised by machine learning models

| Locus | Band | Model | SNP | Gene (Known Locus) |
|---|---|---|---|---|
| 1 | 1q32.2 | A, C | rs6666604,rs1367068* | CR1;CD55 |
| 2 | 2q14.3 | A, B, C | rs79527490,rs3754617* | BIN1 |
| 3 | 4p16.3 | A, C | rs4690324,rs4690221* | IDUA |
| 4 | 5p15.2 | C | rs31930,rs11133794 | OTULIN |
| 5 | 5q14.3 | A, C | rs4141503 | RASA1 |
| 6 | 5q35.3 | A, C | rs113706587 | RASGEF1C |
| 7 | 6p21.1 | C | rs9381040,rs3747742* | TREML2;NFYA |
| 8 | 6p21.32 | A, C | rs3135392,rs6931277* | HLA-DQA2;HLA-DRB5 |
| 9 | 8p21.1 | A, C | rs7831810,rs10866859* | EPHX2;CLU;SCARA3 (CLU) |
| 10 | 8p23.1 | C | rs4731,rs1065712 | FDFT1 |
| 11 | 9q31.1 | B | rs76996768,rs2777800* | ABCA1 |
| 12 | 11q12.2 | A, C | rs983392,rs7232* | MS4A6A;MS4A6E |
| 13 | 11q14.2 | A, C | rs536841,rs471470* | PICALM |
| 14 | 14q32.12 | C | rs10498633,rs12590654 | RIN3 |
| 15 | 15q21.2 | A | rs2306331 | AP4E1 |
| 16 | 19p13.3 | A, B, C | rs11672916,rs3795065* | ABCA7;CNN2 |
| 17 | 19q13.32 | A, B, C, D | rs7412, rs429358* | APOE |
| 18 | 20q13.31 | A, C | rs6069736,rs6127744 | RTF2 (CASS4) |
| 19 | 21q21.3 | A | rs2830500,rs2830520 | ADAMTS1 |

Loci are defined as known if genome-wide significant in major AD GWAS (Supplementary Data 1) and containing at least one SNP in the top variant list from an ML model (A: GBM, B: NN, C: MB-MDRC 1 d, D: MB-MDRC 2 d). Chromosomal bands use build GRCh38p.14 from Ensembl release 111 in BiomaRt. *SNP list curtailed at two rsids: see Supplementary Data 5 for full list.

**Table 2 | 34 putative novel loci prioritised from machine learning models**

| Locus | Band | Model | SNP | Gene | Replication p-value | Summarised GWAS Catalogue Traits |
|---|---|---|---|---|---|---|
| *1* | *2p13.3* | *A* | *rs80275456* | *ARHGAP25* | *0.0116†* | *Blood count, Psychiatric disorders, Lipid metabolism* |
| 2 | 3p21.1 | A | rs3774423 | CACNA1D | 0.7828 | CVD, Sleep, Cognition, Adiposity, Smoking |
| 3 | 3p24.1 | B | rs2371108,rs9828781 | EOMES | 0.6256 | Inflammation, Arthritis, Blood count |
| 4 | 3p25.1 | A | rs7636739 | SH3BP5 | 0.781 | Smoking, Alcohol |
| 5 | 3p25.2 | A | rs7645264 | RPL32 | 0.9424 | Adiposity |
| 6 | 4q21.22 | B | rs17005633 | HNRNPDL | 0.4745 | |
| 7 | 4q27 | A | rs75623120 | TRPC3 | 0.2129 | Inflammation |
| 8 | 4q34.3 | A | rs1363592,rs2546251 | LINC00290 | 0.4379 | AD in APOE-e4, CSF AB1-42 levels, CAA x APOE-e4, Psychiatric disorders |
| 9 | 6p21.33 | A | rs12210887 | LSM2 | 0.933 | Psychiatric disorders, Inflammation, Infection |
| 10 | 6p22.2 | A | rs9393777 | BTN3A1 | 0.6977 | Diabetes, Psychiatric disorders, Cognition, Adiposity |
| 11 | 7p15.3 | A | rs11770728 | CCDC126 | 0.0832 | Liver, Cardiovascular function |
| 12 | 7p21.2 | A | rs10486769,rs16878585 | MEOX2 | 0.0564 | CVD, Lipid metabolism, Brain volume,Adiposity |
| 13 | 8p12 | A | rs76461905,rs11780927 | RNF122;DUSP26 | 0.4174 | Sleep, BMI, Pulse pressure |
| 14 | 8p23.2 | A | rs11136920 | CSMD1;LOC100287015 | 0.4383 | AD, CVD, Inflammation, Sleep, Diabetes, Psychiatric disorders, Smoking, Lipid metabolism |
| 15 | 8p23.3 | A | rs1550948,rs73175035 | FBXO25 | 0.0617 | Inflammation, Alcohol, Moyamoya disease |
| 16 | 8q21.13 | A | rs11778492 | ZC2HC1A | 0.2123 | Depression, Inflammation, Weight |
| 17 | 8q21.13 | A | rs11987678 | TPD52 | 0.6003 | Inflammation, Diabetes, Sleep |
| *18* | *8q24.3* | *A* | *rs7013750* | *LY6H* | *0.0449†* | *Smoking, Diabetes, Cognition* |
| 19 | 13q14.13 | A | rs9567575 | SIAH3 | 0.8082 | NFTs, Clusterin levels, Diabetes, Cortical surface area, Cardiovascular function |
| 20 | 13q33.1 | A | rs1333277 | DAOA-AS1;LINC01309 | 0.0992 | Smoking, Diabetes, Cognition, CSF TREM-2, PHF-tau, Adiposity |
| 21 | 14q11.2 | A | rs1107390 | DAD1;ABHD4 | 0.991 | Inflammation, liver function |
| 22 | 14q32.33 | A | rs7143644 | C14orf180;TMEM179;INF2 | | Inflammation, Psychiatric disorders |
| 23 | 15q21.1 | A | rs498976 | SEMA6D | 0.9661 | Cognition, Psychiatric disorders, Sleep, Longevity, Smoking, Anthropometric traits |
| *24* | *15q26.2* | *A* | *rs117354036,rs4247092* | *LINC00924;LOC105369212* | *0.0345†* | *Cognition in MCI, Inflammation, Smoking, CVD* |
| 25 | 15q26.2 | A | rs12911308 | MCTP2 | 0.0788 | CSF pTau in AD, CVD, Inflammation, Diabetes, Psychiatric disorders |
| 26 | 15q26.3 | A | rs12901450,rs8035839 | TARS3 | 0.7071 | |
| *27* | *16p12.2* | *A* | *rs250583* | *COG7* | *0.001†* | *Cholesterol x fenofibrate* |
| *28* | *16p13.3* | *A* | *rs4786422* | *ZNF597* | *0.0227†* | |
| 29 | 17p13.1 | C | rs11078722 | KCNAB3 | 0.2143 | Smoking, Blood pressure |
| 30 | 18q12.2 | A | rs17561693,rs1941955 | MIR4318;MIR924HG;CELF4 | 0.349 | Smoking, Obesity, Diabetes, Psychiatric disorders, Sleep |
| 31 | 18q21.33 | A | rs694419,rs17729530 | ZCCHC2 | 0.2432 | PHF-Tau, Sleep, Psychiatric disorders, Inflammation, Cognition, Adiposity |
| 32 | 20q11.22 | A | rs910873 | ASIP | 0.942 | AD AAO, Skin cancer, Smoking |
| *33* | *21q22.11* | *B* | *rs4998557,rs4816407** | *SOD1* | *0.0147†* | *ALS, Globulin-levels* |

Top predictors from machine learning models within 500 kb of each other were merged using bedtools v2.27.1, or if functionally or positionally annotated with the same gene. We prioritise the 34 loci which have support from more than one split in a machine learning model (A: GBM, B: NN, C: MB-MDRC 1 d, D: MB-MDRC 2 d) or replication in an external dataset (Jansen et al., 2019[22]), defined as at least one SNP from the listed rsIDs in that row of the table which is present at $p ≤ 0.05$ in publicly available summary statistics. Chromosomal bands were extracted for GRCh38p.14 from Ensembl release 111 using BiomaRt; gene annotation methodology is described in the methods section. The direction of effect is not included here as for ML approaches no single direction is obtained: different subgroups may have varying associations with the outcome for a given SNP. Consequently, direction of effect is not as informative as in standard GWAS replication studies. Traits related to AD and dementia which were annotated from the GWAS catalogue and summarised; a full list of traits from past GWAS and corresponding PubMed IDs is given in Supplementary Data 5. *SNP list curtailed at two rsids: see Supplementary Data 5 for full list. †Nominally significant in the replication. Putative novel loci (emphasised) are discussed in the manuscript.

status (rs7412 and rs429358) identified and ranked as the most important SNPs by both GBMs and NNs but not by a GWAS in the training set (Fig. 3b). To compare the ML findings with GWAS results, we included SNPs not only identified in the training set in our study, but also all SNPs reported as genome-wide significant by larger meta-analyses (Supplementary Data 1) and applied the same gene annotation strategy as for the top variants from ML models (see *"functional annotation and enrichment analysis"*). In total, 130 genes were annotated, where more than one gene may be annotated

to the same locus, and the same gene may be annotated to several independent loci. These genes correspond to 86 distinct loci reported in previous publications[23–25]. Of these, 19 loci were implicated by at least one ML methodology (Supplementary Fig. 8). *APOE* was found by all methods, seven loci (*PICALM, IDUA, RASGEF1C, CLU, CR1, RTF2, RASA1*) were found by GBMs and MB-MDRC 1 d, and two (*ABCA7, BIN1*) by MB-MDRC 1 d, GBMs and NNs. *ABCA1* was only detected by NNs, and two (*SPPL2A, ADAMTS1*) only by GBMs (Table 1).

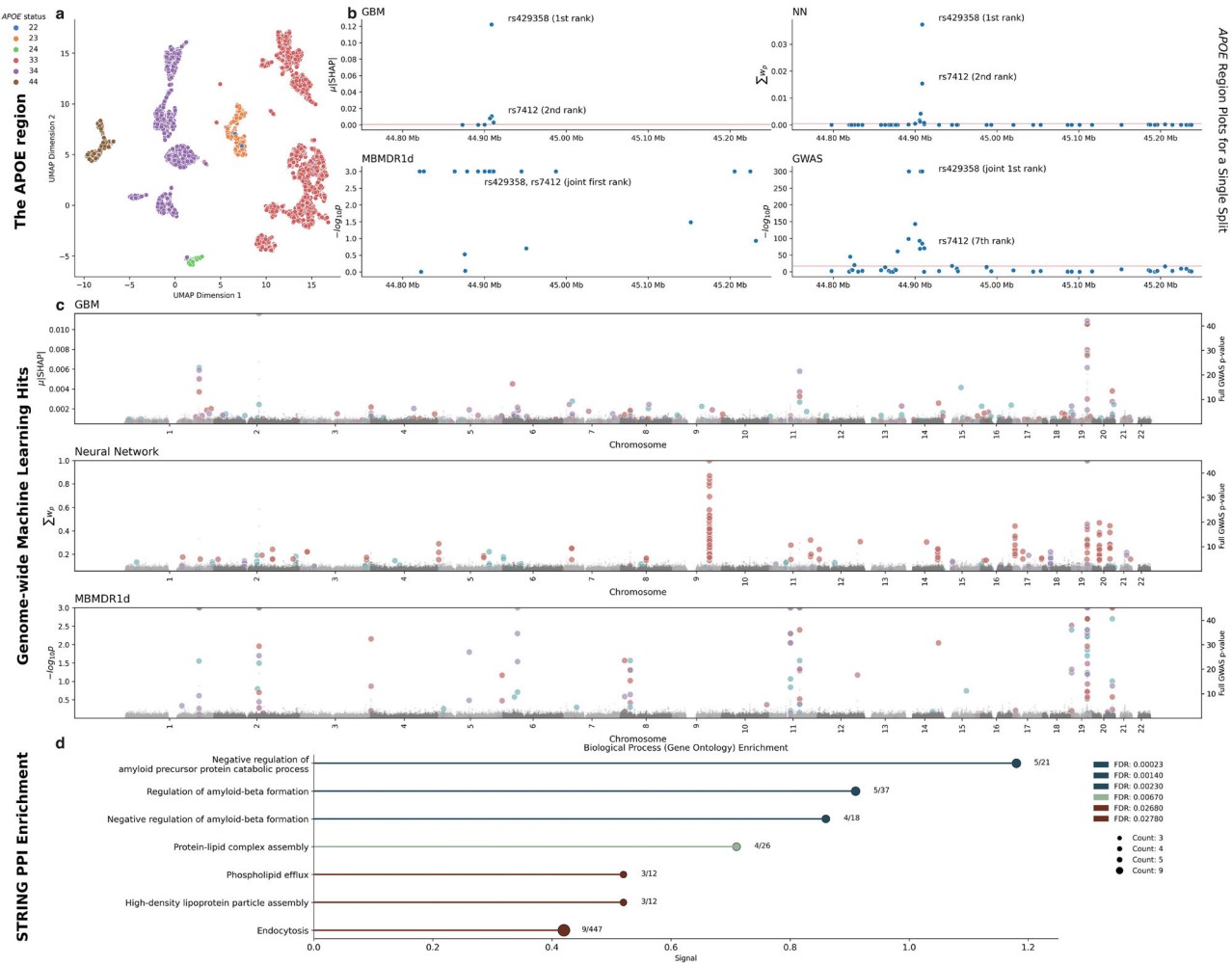

**Fig. 3 | Association in ML models.** Uniform Manifold Approximation and Projection (UMAP) of raw (unscaled) SHAP values for GBM hits highlights that *APOE* alleles are identified and drive prediction (**a**). Neural networks and gradient boosting both rank the SNPs required to derive the e2 and e4 allele status for *APOE* as highest, unlike traditional GWAS (**b**). Values for neural networks and GBMs are not based on *p*-values, as described below, while p-values in (**b**,**c**) (GWAS) are from a logistic regression in the training split, using a logistic regression and *p*-values from a two-sided Wald test as standard. Manhattan plots are given for top hits only from gradient boosting (mean absolute SHAP values), neural networks (normalised network layer weights) and MB-MDRC 1 d (−log10 *p*-values), where hits from different random splits of the models are shown in different colours, and all variants from a single GWAS on the train split are shown in greyscale for comparison (right hand y-axis) (**c**). *p*-values for MB-MDRC 1 d in (**b**, **c**) are derived from a two-sided permutation-based test as implemented in MDMDR[61,62]. Hits from machine learning models (see Table 2) are enrichment for known Alzheimer's disease processes (**d**).

Motivated by published evidence suggesting that different variants are associated with the disease depending on *APOE* status, ML models were compared when trained with and without the *APOE* region in the same train-test split. Remarkably, GBMs without the *APOE* region found more known AD risk genes than when trained with *APOE* (see Supplementary section, 2.6), but with lower AUCs (Supplementary Fig. 9).

When the list of SNPs is limited to those which could be identified in the *current data* (using a GWAS in the corresponding training splits), all GWAS-significant SNPs were prioritised by at least one ML approach (Fig. 4). Furthermore, ~84% at the suggestive significance level ($p \leq 10^{-5}$) were retrieved by at least one ML approach.

**Enrichment of ML findings in biologically relevant gene sets**

The SNPs prioritised by ML (Tables 1 and 2) showed significant enrichment in microglial ($p = 0.0024$) and astrocytic regions ($p = 0.0083$), but not synaptic regions ($p = 0.117$). All 68 genes from 52 loci reported in Tables 1 and 2 were further analysed for protein-protein interaction scores using the online tool STRING (Supplementary

Fig. 10). Pathway analyses demonstrated enrichment in various gene-ontology (GO) pathways (Fig. 3d, Supplementary Data 7). As expected, GO pathways yielded a number of biological processes of interest to Alzheimer's disease, including regulation of amyloid beta formation ($p_{FDR} = 0.0014$) and regulation of amyloid precursor protein catabolic process ($p_{FDR} = 0.00023$, Fig. 3d).

## Discussion

In leveraging the largest genotyped case-control AD dataset, this study demonstrates that the current scale of data has reached a threshold where ML can achieve similar predictive accuracy to classical methods and uncover novel genetic insights into AD. Gradient boosting, neural networks, and multifactor dimensionality reduction were selected due to their prominence in the field, demonstrated performance, and complementary strengths. These methods offer distinct advantages: highly performant tree ensembles (GBMs), flexible networks incorporating prior knowledge (NNs), and detection of SNP-SNP interactions (MB-MDR)[26]. Here they were applied to identify novel genes not detectable via GWAS, and to compare ML prediction accuracies with PRS.

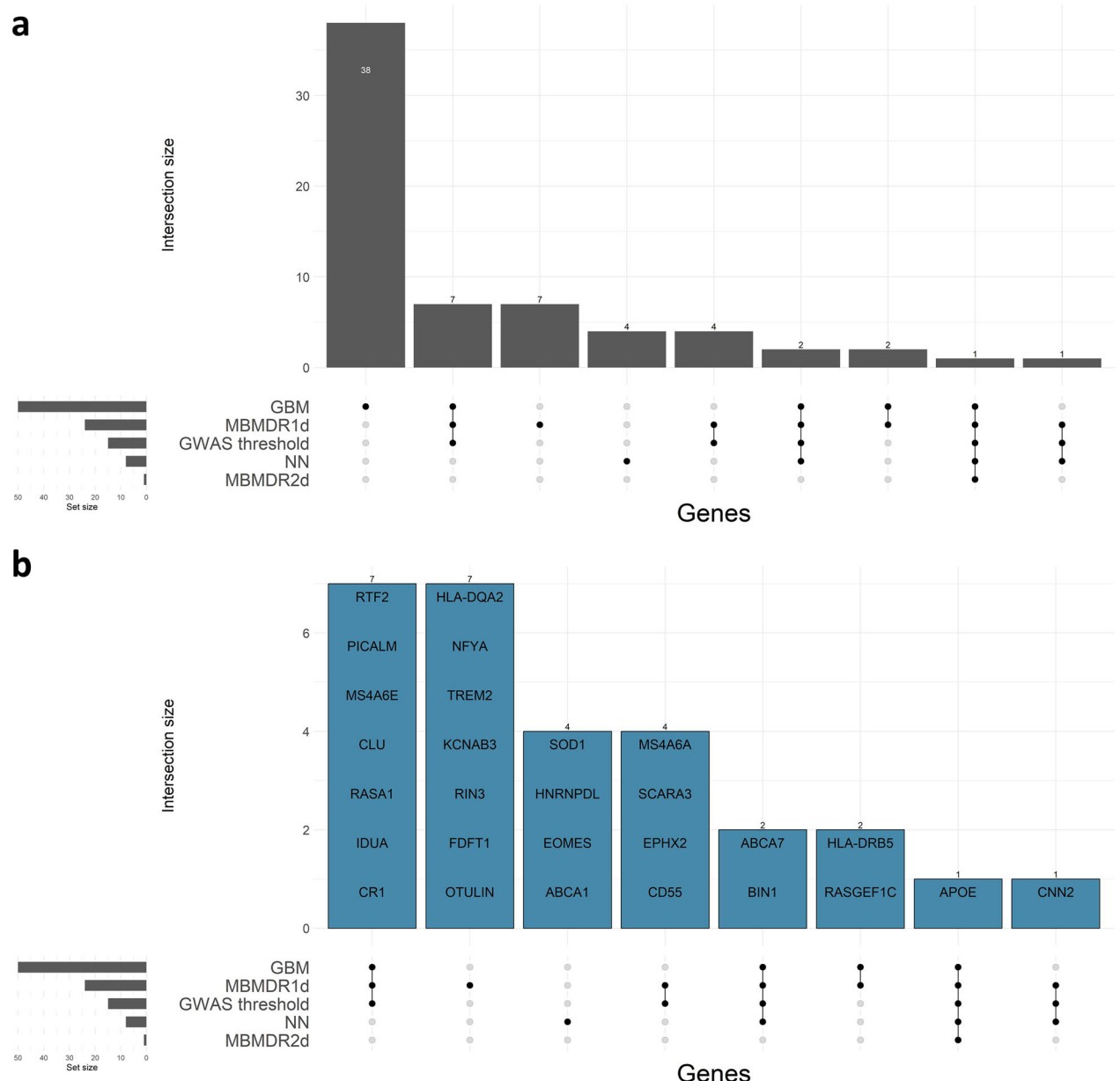

**Fig. 4 | UpSet plot showing the overlap between ML and GWAS significant findings from the train part of the train-test split. a** Genes mapped by the SNPs highlighted by each ML approach separately. Both ML approaches and GWAS significant SNPs were identified in the same training split. **b** Genes that are shared among at least two train-test splits. All GWAS genome-wide significant ($p \leq 5 \times 10^{-8}$) SNPs in the train split were also identified by ML approaches.

For simplicity, genes within 500 kb of a known locus or with at least one overlapping gene with the region were annotated only by the locus, including *MS4A6A, CSTF1, EPHX2, CNN2, TOMM40/NECTIN2/CLPTM1/BCL3/BCAM/ APOC1/APOC2/APOC4* and *DGKQ/FAM53A*, which were mapped to *MS4A\*, CASS4, CLU, ABCA7, APOE* and *IDUA* regions, respectively. Subplots were created using ComplexUpset version 1.3.3.

ML methods correctly identified the lead SNPs used to calculate the *APOE* haplotype, rs7412 and rs429358, as having the greatest impact on the model, in contrast to classical univariable linear models (traditional GWAS) which do not distinguish between top variants by *p*-value, and rank rs7412 lower (Fig. 3c). PRS for AD risk prediction perform best when modelled with two predictors: PRS calculated without the *APOE* region and a separately coded weighting of the *APOE* e2 and e4 alleles[27]. Here, we demonstrate that ML approaches can accurately identify *APOE* clusters and achieve comparable prediction accuracy without including the derived *APOE* variable as a second predictor. ML approaches further detected the lead SNPs identified in large meta-analyses GWAS for several key genes, including *BIN1* (rs6733839), *PICALM* (rs3851179), *ABCA7* (rs3752246, a coding variant),

*CASS4* (rs6069736) and *CR1* (rs4844610). More broadly, ML highlighted SNPs in around 22% of genes identified by larger meta-analysis GWAS[23–25], while comparison to a GWAS undertaken on the same (EADB-core) split that was used to train machine learning models, showed that 100% of genome-wide significant SNPs ($p \leq 5 \times 10^{-8}$) were retrieved by ML approaches. This demonstrates that the majority of findings expected under a linear additive GWAS paradigm can be prioritised using flexible machine learning models. Though strides have been made in ML-based prediction of AD from genetic data, for example[28], to our knowledge we present the first well-powered, genome-wide ML-based gene discovery study detecting nearly a quarter of known genes found in larger GWAS meta-analyses from the literature which contain around twenty times the sample size, while

also identifying multiple putative regions with credible associations to AD biology, marking an important benchmark for these methods.

Putative novel genes which replicate in an independent GWAS consist of *ARHGAP25, COG7, LINC00924/LOC105369212, LY6H, SOD1* and *ZNF597*, which have potential relevance to AD. *ARHGAP25* encodes the Arhgap25 protein which is expressed in macrophages where it affects phagocytosis[29] through modulation of the actin cytoskeleton[30]. Wu et al.[31] demonstrated that Ly6h is among the proteins competing to bind to α7 subunits of nicotinic acetylcholine receptors (nAChRs), which are expressed throughout the brain and enable fast cholinergic transmission at synapses. These proteins function collectively to maintain the optimal α7 assembly required for neuronal function and viability. This delicate balance is disrupted during Alzheimer's disease due to Aβ-driven reduction in Ly6h. Notably, an increase in Ly6h in human cerebrospinal fluid correlates with elevated Alzheimer's disease severity[31].

*COG7* (Component of Oligomeric Golgi Complex 7) encodes a protein integral to Golgi apparatus function, which is essential for protein glycosylation and trafficking. Disruptions in Golgi function have been implicated in various neurodegenerative disorders, including Alzheimer's disease. Mutations in COG7 are associated with Congenital Disorders of Glycosylation (CDG)[32], which frequently manifest with neurological impairments. Abnormal glycosylation processes have been implicated in Alzheimer's disease pathology, particularly affecting tau protein processing and amyloid precursor protein (APP) metabolism[33].

ML hits which replicate also include the missense variant rs2306331 (*AP4E1*). This is within the known locus of *SPPL2A* (maximum $r^2$ 0.32, minimum distance 177 kb with rs2306331 and GWAS catalogue hits for *SPPL2A*), but may be an independent signal with the region. The *AP4E1* gene encodes the ε subunit of adaptor protein complex-4 (AP-4), which facilitates the transport of amyloid precursor protein (APP) from the trans-Golgi network to endosomes. Disruption of the APP-AP-4 interaction enhances γ-secretase-catalysed cleavage of APP to amyloid-β peptide, suggesting that AP-4 deficiency may constitute a potential risk factor for Alzheimer's disease[34].

We also note that ML-derived hits in the know loci *IDUA* and *DGKQ* (rs4690324, linked via sQTLs in GTEX to splicing of *IDUA* in the brain[35]) are also related to heparan sulfate metabolism[36], in addition to multiple brain diseases[37], traits[38], and lipid metabolism[39]. Neural networks also highlight a new potential AD-relevant locus. All GO-derived pathways, independent of association to AD, were used in the neural network analysis as hidden layers. This approach highlighted SNPs mapping to *SOD1* (cytogenic band 21q22.11), found in the region of *APP* (21q21.3), with lead SNPs from NNs around 5 Mb away from the closest NN-based hits in *APP*. *SOD1* has been widely investigated for its role in antioxidant defense system, showing impaired expression in AD patients[40,41], but is currently primarily implicated in ALS.

In our study, we adopted an MB-MDR approach to detect both SNP main effects and pair-wise SNP-SNP interactions. The method is an improvement over classical pair-wise statistical interaction approaches as it can detect more complex interactions. However, it implements a somewhat conservative strategy for variant discovery, as interaction studies are hampered by several factors that can increase the number of false positives including, but not limited to, LD and interference of major loci, which may contribute to phantom epistasis. To address this limitation, we opted for a model-based (MB) form of MDR, though the results did not highlight novel loci.

Comparing individual scores, ML predictions achieved correlations of 0.8-0.84 with PRS, indicating ML models give predictions which are broadly consistent with well-established approaches. It also shows that identification of novel signals through flexible modelling of complex effects will introduce deviations from the predictions of a simple linear additive model. Disease risk prediction from ML, as assessed by AUC, was higher (but not significantly) than PRS. Similar

results have been reported in psychiatry[21] and coronary artery disease[42]. This is likely to be due to several reasons. First, SNPs in general are (at best) only correlated with the causal variants, making it particularly difficult to detect nonlinear effects and interactions, which are the main potential advantages of ML over PRS. Second, genetic predictors are weak as compared to some other predictors (e.g. biomarkers[43]), and the upper bound for AUC in complex trait genetics in practice falls substantially below 1[44]. Weak predictor-response relationships are an inherent challenge to finding patterns with flexible models, and complex models may at-present still be under-powered to achieve a clear improvement in AUC. Third, large GWAS discover and replicate SNPs, which in the resulting summary statistics show small association effect sizes. The effect sizes may however be higher in more homogenous samples. For example, the OR for *APOE* is around 3.4 in samples of mean age 72–73 years[45] but is reduced in samples over 90 years[46]. In pathologically-confirmed samples, which are also generally older than clinical samples, some of the GWAS-derived SNP effect sizes are higher than reported in clinically-assessed AD GWAS[47]. In more homogenous samples in terms of age, population, and cognitive scores (such as The *Alzheimer's Disease Neuroimaging Initiative (ADNI)*[48]), the AD PRS AUC are higher than that in clinical samples[49,50]. Thus, summary statistics from large GWAS meta-analyses enable PRS which predict with moderate AUC in any data set, but which do not achieve high accuracy as effect sizes are averaged across many studies with slightly different features such as recruitment criteria and outcome definitions, and therefore genetic architectures, rather than being specific to a particular one.

A similar situation affects variant discovery from flexible models which can detect interactions: nuanced relationships between predictors are notoriously difficult to replicate[51], a situation further complicated by varying LD between tagged and causal variants. Such SNPs may nonetheless represent important loci which impact disease risk in specific contexts without being consistently associated across enough studies and circumstances to reach genome-wide significance under linear models in meta-analyses. While machine learning models here do not show signs of overfitting, and top SNP rankings by SHAP values are consistent in both the train and test splits, many SNPs identified do not replicate in an external dataset, which is also the case for standard GWAS. In particular, in PRS approaches using different priors (for Bayesian models) or LD pruning parameters (in clumping and thresholding approaches), the resulting set of SNPs and their estimated effects will differ[27,52]. Similarly, when effects of SNPs are jointly estimated in sparse, high-dimensional ML models, either simultaneously or in a stage-wise manner (as in gradient boosting), different top associations and predictions are expected. Despite these caveats, we show that the novel findings suggest SNPs which are biologically relevant to AD.

This study has a number of limitations. First, our study attempted to run and compare reasonably diverse methods for predicting AD risk, with the advantage of implementing them in a unified dataset. As a result, we found that the PRS and ML prediction showed a similar prediction performance based upon ranking individuals according by their prediction scores. The predictions from ML and PRS, however, were highly correlated, explaining the similarity of the prediction accuracies and reducing the likelihood that an ensemble of the different methods would improve prediction. Second, the methods used for selection of top SNPs in interpreting model results reflect inherent differences in how the ML approaches work, and no standard thresholds exist at the moment for identifying important features across these distinct frameworks. This methodological variability, which is a broader challenge in the field, likely contributes to the incomplete overlap between the top SNPs identified by different ML approaches, making replication in external summary statistics a critical step to ensure robust findings. Third, the influence of *APOE* status on model outcomes is a key component of all models in the study. While *APOE*

SNPs were included as predictors, we did not conduct analyses stratified by *APOE* carrier status, especially ε3/ε3, ε3/ε4, or ε4/ε4 carriers. Future work should explore whether predictive accuracy and associations differ meaningfully across these strata. Finally, although our post-hoc analyses indicate that cohort or genotyping centre effects do not drive associations or predictions, alternative machine learning approaches such as federated learning (FL) may improve modelling where underlying predictor distributions diverge[53], while also allowing for models which include data from more cohorts without diminishing privacy. Indeed, within a central learning paradigm, current best practices for data harmonisation involve merging datasets based on shared high-quality SNPs that have undergone rigorous quality control, ensuring high imputation accuracy, reasonably large minor allele frequencies (MAF), and other metrics in each separate study. However, these QC steps alone do not guarantee that differences arising from ancestry, clinical assessment criteria, and genotyping chips or batches used to generate the data are fully accounted for. Applying flexible ML within an FL framework leverages the advantages of client-specific data, which is often more homogeneous as it is typically generated locally, within the same population, using similar clinical assessments and the same genotyping chip. Results from each "client" can then inform analyses for other clients by tuning parameters (e.g., increasing or decreasing neural network weights for variants in specific genomic regions), creating a more flexible and adaptive analysis framework.

In conclusion, this study demonstrates that machine learning can uncover both known and novel genetic loci for AD, providing a powerful complement to traditional GWAS while still achieving competitive predictive performance. Though replication challenges remain, ML successfully prioritised established risk loci and identified biologically relevant novel associations, including variants in *ARHGAP25*, *COG7*, *LY6H*, and *SOD1*. These findings highlight ML's potential to refine our understanding of AD beyond additive genetic effects and expand the toolkit available for maximising discovery from available data. With expanding datasets and computational advances, ML could further enhance risk prediction and gene discovery, particularly through federated learning and multi-omic integration. This study marks a key step toward leveraging ML for deeper insights into AD genetics.

## Methods

### Data
Data were obtained from the European Alzheimer & Dementia Biobank (EADB) consortium which combines genetic and clinical data from 16 countries and has been described previously[23]. All study protocols were reviewed and approved by the respective institutional review boards overseeing the cohorts (see supplementary information for details). Individuals were genotyped at three centres. Data were accessed after quality control procedures and data harmonisation were applied to give the EADB-core sample. All participants are unrelated individuals of European ancestry, encompassing 20,013 clinically defined AD cases and 21,673 control, after excluding participants present in Kunkle et al.[45]. 59% of the sample is female, split as 62% in cases and 57% in controls, with a median age at baseline of 73 (IQR = 14). Data and splitting procedures are described further in Supplementary section 1.1. Informed consent was obtained in writing from all study participants. For individuals with significant cognitive impairment, consent was secured from a caregiver, legal guardian, or other authorized proxy.

### Quality control (QC)
Analyses used directly genotyped (non-imputed) variants to ensure high data quality. To avoid excluding key known AD loci not covered in the genotyped data, we also included 67 imputed variants in previously reported genome-wide significant loci[23–25] (Supplementary Fig. 1) which were not already present in the genotyped data, converting the imputed dosages to the most probable (probability ≥ 0.9)

genotypes, and applying further quality control as described previously[23]. Data were further filtered for a minor allele frequency of 5%, to ensure variants were common enough to reliably observe interactions. SNPs were clumped ($R^2 = 0.75$ following Joiret et al.[54], window = 1 Mb) using stage 1 summary statistics from Kunkle et al.[45], after removing individuals common between the summary statistics and the genotyped data. Out of 81 SNPs previously reported as genome-wide significant, 52 survived minor allele frequency (MAF) and clumping procedures (Supplementary Fig. 1; Supplementary Data 1), giving a combined 215,193 SNPs (Supplementary section 1.1). Analyses were also run without inclusion of the previously reported 67 imputed variants to confirm that their inclusion did not artificially inflate performance estimates.

### Statistics and reproducibility
No statistical method was used to predetermine sample size, as there are no standardised calculations for machine learning. For consistency, all implemented approaches were evaluated using the same QC and random train-test splits of the data (Fig. 1) which were well balanced for case-control status, age-at-baseline or assessment, sex, genotyping centre, and the distribution of all principal components (Supplementary Fig. 2). Participants were randomly separated into 70–30% train-test splits, with the same split applied to all algorithms, resulting in 29,180 individuals in the training set (14,006 cases; 15,174 controls) and 12,506 for testing (6007 cases; 6499 controls), each with 215,193 predictors after quality control procedures. ML models were built with and without SNPs from the *APOE* region (Chr19:44.4–46.5 Mb). In training ML and PRS models, analyses were adjusted for covariates comprising genetic sex, 20 principal component (PCs), and genotyping centre. The adjustment method was altered to be appropriate for each modelling approach: covariates were included in the final layer for NNs, with predictions and importance scores taken from non-covariate nodes (see Supplementary Fig. 3); for GBMs and MB-MDR, covariates were z-transformed and then regressed-off from the data before modelling. All reported area under the receiver operator characteristic curve (AUC) values are calculated on the predicted probabilities from models (without thresholding) and adjusted again for confounders in the test split. We utilise penalisation and cross-validated random (GBMs) or grid-based (NNs and MB-MDR) hyperparameter search to reduce the likelihood of overfitting. Details for training and covariate adjustment are given in Supplementary sections 1.2-1.4. To ensure robust results, stability was assessed by re-running all models on three additional random 70–30% train-test splits of the data.

### Gradient boosting machines (GBMs)
GBMs were trained using version 1.7.6 of the XGBoost package[55], an efficient implementation of regularised gradient boosting, and the dask package[56], version 2023.1.1, which allows for distributed training of ML models across multiple nodes in a high-performance computing (HPC) cluster. Hyperparameters for learning rate, tree depth, and column sampling fraction were tuned on a random subsample of the training set using random search (Supplementary section 1.2). Importance of all predictors in GBMs was assessed using SHapley Additive exPlanatory (SHAP) values[57,58].

### Neural networks (NNs)
NN models were built with GenNet[59], which uses a biologically-driven configuration in which the connections between the input layer, representing SNPs, and the first hidden layer, representing genes, are defined using knowledge available in annotation databases. NN architectures were built with nine hidden layers, including an initial SNP-to-gene layer annotating all SNPs to the nearest gene using annotations from ANNOVAR[60], followed by layers defined from the hierarchy of terms from the Gene Ontology consortium (GO terms), where

connections between layers progress from local pathways to more general ones as they move deeper through the network, using all available pathway annotations (Supplementary Fig. 3, Supplementary Data 2). Model hyperparameters for batch size, learning rate (LR) and $L_1$ (default) penalization were tuned during training (Supplementary section 1.3).

## Model based multifactor dimensionality reduction (MB-MDR)

A multi-dimensional reduction strategy was implemented using the MB-MDR methodology[61,62] with MBMDR 4.4.1 software. An approximation routine was used to accelerate permutation-based significance assessment and multiple testing[63], when searching for disease-susceptibility multi-locus genotypes, adjusted for confounders. Single and interacting variants under various pair-wise and higher order epistasis models were combined to create multilocus risk scores (MB-MDRC) and estimate an individual's susceptibility to a trait with the MBMDRClassifieR package available in R[64]. SNPs and SNP-SNP interactions were included in the MB-MDR risk score if they passed the permutation-based multiple testing corrected threshold that had the best performance in the train split (Supplementary section 1.4).

## Polygenic risk scores (PRS)

PRS were calculated with LDAK-Bolt-Predict from the LDAK package, the most predictive heritability and linkage disequilibrium-informed PRS when individual genotypes are available[65]. LDAK-Bolt-Predict reweights variants using Gaussian priors informed by heritability models, shrinking effect sizes without the need for $p$-value thresholding[65]. To give the same information to both PRS and all machine learning models, PRS were derived using summary statistics which were generated in the training set by running a genome-wide association study (GWAS), adjusting for the same confounders in PLINK v2.00a3.3LM[66] (Supplementary section 1.5).

## Selection of top predictors

The top SNPs identified by machine learning were selected by the permutation-based $p$-value threshold defined above for MB-MDR ($p_{adj} < 1$), empirically by taking the extreme tail of the distribution for SHAP values ($\mu | \text{SHAP} | > 0.0005$) in gradient boosting (Supplementary Fig. 4), summed weights ($p_{adj} < 0.05$) for neural networks (Supplementary Fig. 5), and by applying the Boruta algorithm[67] (gradient boosting models only). Only SNPs which were present in top SNPs for at least two train-test random data splits of a given method were prioritised. Details on predictor selection are given in Supplementary sections 1.6 and 2.2-2.4.

## Replication of top predictors

The prioritised loci were reported as novel if they were identified with more than one split of a machine learning model, replicated at $p \leq 0.05$ significance level in an external independent dataset (Jansen et al.[22]) and had no genome-wide hits from AD summary statistics in the GWAS catalogue. The direction of effect for ML approaches is not available as different subgroups may have varying associations with the outcome for a given SNP, and consequently it is not directly reportable as in standard GWAS.

## Functional annotation and enrichment analysis

Publicly available tools were used for further annotation: SNPs were annotated with dbSNP build 156[68]. SNPs were initially positionally mapped to genes using ANNOVAR[60] version 2020-06-07 and Gencode v40, and then reannotated using functional evidence from the Open Targets Genetics portal where available[69]. Genome coordinates use build GRCh38.p14. Pathway analysis and protein-protein interaction from the consensus annotated genes were determined using STRING v12.0[70] (Supplementary section 1.7).

The list of top SNPs within each ML analysis was tested for enrichment in the list of genes expressed in microglia ($n = 761$)[71], astrocytes ($n = 757$)[72], and synapses ($n = 1535$)[73], with a window size of 35 kilobase (kb) upstream and 10 kb downstream of regions to include regulatory elements. To account for non-independent SNPs in the genomic regions, enrichment $p$-values were derived using a bootstrap approach (Supplementary section 1.8) and presented without correction for multiple testing for the number of cell types tested.

## Statistical interactions

The ML methods used may include interactions but not explicitly test for them. The top SNPs from all ML approaches (Tables 1 and 2) were therefore formally tested for pair-wise interactions in the whole dataset under a regression framework, assuming a multiplicative relationship between SNPs, i.e. $\text{logit}(y) = \beta_0 + \beta_1 \text{SNP}_1 + \beta_2 \text{SNP}_2 + \beta_3 \text{SNP}_1 * \text{SNP}_2$, where SNPs are coded additively (0, 1, 2) with zero-count compensation, and covariates are included in the model. Pairs were assessed by the $p$-value for the interaction term after adjusting for multiple comparisons using a false discovery rate (FDR) Benjamini-Hochberg ($p = 0.05$) threshold for significance (Supplementary section 1.9). Putative SNP interactions were visualized in python using shap 0.41[58], statsmodels 0.13.5[74], matplotlib 3.7.1[75], and seaborn 0.12.2[76]. The overall workflow is presented in Fig. 1.

## Reporting summary

Further information on research design is available in the Nature Portfolio Reporting Summary linked to this article.

# Data availability

The raw data are protected and are not publically available due to data privacy laws. They can be accessed by application to the EADB consortium. The data generated in this study are provided in the Supplementary Information file.

# Code availability

Code for running biologically-informed neural networks via GenNet and Tensorflow is already published and available (https://github.com/ArnovanHilten/GenNet). Code used for running distributed gradient boosting using dask and XGBoost (DAXOS v0.1.0) with covariate adjustment, preprocessing of genotypes, and applying distributed Boruta-SHAP feature selection has been made publicly available (https://github.com/seafloor/daxos)[77]. Pathway annotation and enrichment tests are also available on GitHub (https://github.com/seafloor/escott-price-lab-pipelines/tree/main/workflows/pathways). We use MBMDR version 4.4.1 (http://bio3.giga.ulg.ac.be/index.php/software/mb-mdr).

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

## Acknowledgements

We thank the University of Lille's Intensive Scientific Computing Mesocentre. M.B.S. and V.E.P. are supported by the UK Dementia Research Institute [UK DRI-3206] through UK DRI Ltd, principally funded by the Medical Research Council and by the UK Medical Research Council: MR/T04604X/1, MR/P005748/1. K.V.S. has received funding from the European Union's Horizon 2020 research and innovation programme under the Marie Sklodowska-Curie grant agreements No 813533 (MLFPM) and No 860895 (TranSYS), and FNRS convention PDR T.0294.24 "Expanded PRS embracing pathways and interactions for increased clinical utility". F.M. (Federico Melograna) was also funded by the Marie Sklodowska-Curie grant agreement 860895 (TranSYS). J.W. (Julie Williams) is supported by Moondance Dementia Research Laboratory. B.M.T. and G.R. (Gennady Roshchupkin) are funded by ZonMw VIDI (#09150171910068) and ZonMw Veni grant (1936320) respectively. N.A. is funded by GSK. RG and AD were supported by the Italian Ministry of Health (Ricerca Corrente).

## Author contributions

M.B.S., F.M. (Federico Melograna) and B.U. contributed equally to the study conception and design, analysis and interpretation of data, creation of code for analysis and primary drafting and revision of the manuscript. C.B., B.G.B., D.D., A.J.N., P.H., B.M.T., M.H. (Marc Hulsman), I.D.R., R.C.M., A.R. (Agustín Ruiz), R.F.S., N.A., G.R. (Gennady Roshchupkin) and J.C. contributed to the study design, interpretation of data or substantial revisions. K.V.S., C.V.D. and V.E.P. contributed equally to the study conception, interpretation of data, manuscript drafting and substantial revisions. All remaining authors—S.V.D.L., A.C., F.K., O.P., A.S. (Anja Schneider), M.D., D.R., N.S. (Norbert Scherbaum), J.D., S.R.H., L.H., L.M.P., E.D., T.G., J.W. (Jens Wiltfang), S.H.H., S.M. (Susanne Moebus), T.T., N.S. (Nikolaos Scarmeas), O.D.I., F.M. (Fermin Moreno), J.P.T., M.J.B., P.P., R.S.V., V.A., M.B., P.G.G., R.P., P.M. (Pablo Mir), L.M.R., G.P.R., J.M.G.A., E.R.R., H.S., S.H., A.D.M., S.M. (Shima Mehrabian), L.T. (Latchezar Traykov), J.H., M.V., N.S. (Nicolai Sandau), J.Q.T., Y.A.P., H.H., J.V.S., I.R. (Inez Ramakers), F.V., P.S., C.G., G.P., V.G., J.W. (Julie Williams), P.A., A.B., J.F.D., G.N., C.D., F.P. (Florence Pasquier), O.H., S.D., E.G., J.P., R.G., D.G., B.A., P.M. (Patrizia Mecocci), V.S., L.P., A.S. (Alessio Squassina), L.T. (Lucio Tremolizzo), B.B., M.W., B.N., M.S., D.S., I.R. (Innocenzo Rainero), A.D., F.P. (Fabrizio Piras), C.M., G.R., F.J., P.K., T.M., P.S.J., K.S., M.I., M.H. (Mikko Hiltunen), R.S., W.V.D.F., O.A.A., A.R. (Alfredo Ramirez), and those in the EADB umbrella group (see

Supplementary)—contributed to data acquisition and/or minor revisions.

## Competing interests

Outside the submitted work, T.G. received consulting fees from AbbVie, Alector, Anavex, Biogen, BMS; Cogthera, Eli Lilly, Functional Neuromodulation, Grifols, Iqvia, Janssen, Noselab, Novo Nordisk, NuiCare, Orphanzyme, Roche Diagnostics, Roche Pharma, UCB, and Vivoryon; lecture fees from Biogen, Eisai, Grifols, Medical Tribune, Novo Nordisk, Roche Pharma, Schwabe, and Synlab; and has received grants to his institution from Biogen, Eisai, and Roche Diagnostics. N.A. received funding from GSK. All other authors declare no competing interests.

## Additional information

**Peer review information** *Nature Communications* thanks Payam Barnaghi, who co-reviewed with Antigone Fogel, Mike Nalls, and the other,

anonymous, reviewer(s) for their contribution to the peer review of this work. A peer review file is available.

Matthew Bracher-Smith [1,2,267], Federico Melograna[3,7,267], Brittany Ulm[4,5,267], Céline Bellenguez [6], Benjamin Grenier-Boley[6], Diane Duroux[7], Alejo J. Nevado[5], Peter Holmans [2], Betty M. Tijms [8,9], Marc Hulsman [8,9], Itziar de Rojas [10,11], Rafael Campos-Martin [12,13], Sven van der Lee [8,9], Atahualpa Castillo[2], Fahri Küçükali [14,15], Oliver Peters [16,17], Anja Schneider [18,19], Martin Dichgans [20,21,22], Dan Rujescu [23], Norbert Scherbaum [24], Jürgen Deckert[25], Steffi Riedel-Heller [26], Lucrezia Hausner [27], Laura Molina-Porcel [28,29], Emrah Düzel[30,31], Timo Grimmer[32], Jens Wiltfang [33,34,35], Stefanie Heilmann-Heimbach [36], Susanne Moebus [37], Thomas Tegos[38], Nikolaos Scarmeas[39,40], Oriol Dols-Icardo [41,42], Fermin Moreno [42,43,44], Jordi Pérez-Tur [42,45], María J. Bullido [42,46,47,48], Pau Pastor [49,50], Raquel Sánchez-Valle[51], Victoria Álvarez [52,53], Mercè Boada [42,54], Pablo García-González [55], Raquel Puerta [55], Pablo Mir [42,56], Luis M. Real [57,58], Gerard Piñol-Ripoll[59,60], Jose María García-Alberca [42,58,61], Eloy Rodriguez-Rodriguez [42,62], Hilkka Soininen [63], Sami Heikkinen [64], Alexandre de Mendonça[65], Shima Mehrabian[66], Latchezar Traykov[67], Jakub Hort [68], Martin Vyhnalek[69,70], Nicolai Sandau[71], Jesper Qvist Thomassen [71], Yolande A. L. Pijnenburg[8], Henne Holstege [8,72], John van Swieten[73], Inez Ramakers[74], Frans Verhey[74], Philip Scheltens [8], Caroline Graff[75], Goran Papenberg[76], Vilmantas Giedraitis [77], Julie Williams [1,2], Philippe Amouyel [6], Anne Boland [78], Jean-François Deleuze[78], Gael Nicolas [79], Carole Dufouil [80,81], Florence Pasquier [82], Olivier Hanon[83], Stéphanie Debette [84,85], Edna Grünblatt [86,87,88], Julius Popp[89,90,91], Roberta Ghidoni [92], Daniela Galimberti [93,94], Beatrice Arosio [95,96], Patrizia Mecocci [97,98], Vincenzo Solfrizzi[99], Lucilla Parnetti [100], Alessio Squassina [101], Lucio Tremolizzo[102], Barbara Borroni [92,103], Michael Wagner [18,104], Benedetta Nacmias [105,106], Marco Spallazzi[107], Davide Seripa[108], Innocenzo Rainero[109], Antonio Daniele [110,111], Fabrizio Piras [112], Carlo Masullo[113], Giacomina Rossi [114], Frank Jessen[18,115,116], Patrick Kehoe [117], Tsolaki Magda [38,118], Pascual Sánchez-Juan[42,119], Kristel Sleegers [14,15], Martin Ingelsson [120,121,122], Mikko Hiltunen[64], Rebecca Sims [2], Wiesje van der Flier [8], Ole A. Andreassen [123,124], Agustín Ruiz [42,125,126], Alfredo Ramirez [12,18,104,127,128], EADB*, Ruth Frikke-Schmidt [71,129], Najaf Amin [4], Gennady Roshchupkin [130,131], Jean-Charles Lambert [6], Kristel Van Steen [3,7,268] ✉, Cornelia van Duijn [4,5,132,268] ✉ & Valentina Escott-Price [1,2,268] ✉

[1]School of Medicine, Cardiff University, Cardiff, UK. [2]Centre for Neuropsychiatric Genetics and Genomics, Division of Psychological Medicine and Clinical Neuroscience, School of Medicine, Cardiff University, Cardiff, UK. [3]BIO3 - Systems Medicine, Department of Human Genetics, KU Leuven, Leuven, Belgium. [4]Nuffield Department of Population Health, University of Oxford, Oxford, United Kingdom. [5]Centre for Artificial Intelligence in Precision Medicine, University of Oxford, Oxford, United Kingdom. [6]Univ. Lille, Inserm, CHU Lille, Institut Pasteur de Lille, LabEx DISTALZ - U1167-RID-AGE Facteurs de risque et déterminants moléculaires des maladies liées au vieillissement, Lille, France. [7]BIO3 - Systems Genetics, GIGA-R Molecular and Computational Biology, University of Liege, Liege, Belgium. [8]Alzheimer Center Amsterdam, Department of Neurology, Amsterdam Neuroscience, Vrije Universiteit Amsterdam, Amsterdam UMC, Amsterdam, The Netherlands. [9]Department of Complex Trait Genetics, Center for Neurogenomics and Cognitive Research, Amsterdam Neuroscience, Vrije

University, Amsterdam, The Netherlands. [10]Research Center and Memory Clinic, ACE Alzheimer Center Barcelona. Universitat Internacional de Catalunya, Catalunya, Spain. [11]Networking Research Center on Neurodegenerative Diseases (CIBERNED), Instituto de Salud Carlos III, Madrid, Spain. [12]Division of Neurogenetics and Molecular Psychiatry, Department of Psychiatry and Psychotherapy, Faculty of Medicine and University Hospital Cologne, University of Cologne, Cologne, Germany. [13]Estudios en Neurociencias y Sistemas Complejos (ENyS) CONICET-HEC-UNAJ, Buenos Aires, Argentina. [14]Complex Genetics of Alzheimer's Disease Group, VIB Center for Molecular Neurology, VIB, Antwerp, Belgium. [15]Department of Biomedical Sciences, University of Antwerp, Antwerp, Belgium. [16]German Center for Neurodegenerative Diseases (DZNE), Berlin, Germany. [17]Charité – Universitätsmedizin Berlin, corporate member of Freie Universität Berlin, Humboldt-Universität zu Berlin, and Berlin Institute of Health, Institute of Psychiatry and Psychotherapy, Hindenburgdamm 30, 12203 Berlin, Germany. [18]German Center for Neurodegenerative Diseases (DZNE), Bonn, Germany. [19]Department of Cognitive Disorders and Old Age Psychiatry, University Hospital Bonn, Venusberg-Campus 1, 53127 Bonn, Germany. [20]Institute for Stroke and Dementia Research (ISD), University Hospital, LMU Munich, Munich, Germany. [21]German Center for Neurodegenerative Diseases (DZNE), Munich, Germany. [22]Munich Cluster for Systems Neurology (SyNergy), Munich, Germany. [23]Department of Psychiatry and Psychotherapy, Comprehensive Center for Clinical Neurosciences and Mental Health (C3NMH), Medical University of Vienna, Vienna, Austria. [24]Department of Psychiatry and Psychotherapy, LVR-Klinikum Essen, University of Duisburg-Essen, Germany Medical Faculty, Essen, Germany. [25]Department of Psychiatry, Psychosomatics and Psychotherapy, Center of Mental Health, University Hospital of Würzburg, Würzburg, Germany. [26]Institute of Social Medicine, Occupational Health and Public Health, University of Leipzig, 04103 Leipzig, Germany. [27]Department of Geriatric Psychiatry, Central Institute for Mental Health Mannheim, Faculty Mannheim, University of Heidelberg, Heidelberg, Germany. [28]Alzheimer's Disease and Other Cognitive Disorders Unit, Neurology Service, Hospital Clínic of Barcelona, Fundació Recerca Clinic Barcelona- Institut d'Investigacions Biomèdiques August Pi i Sunyer (FRCB-IDIBAPS), and University of Barcelona, Barcelona, Spain. [29]Neurological Tissue Bank - Biobank, Hospital Clinic-FRCB-IDIBAPS, Barcelona, Spain. [30]German Center for Neurodegenerative Diseases (DZNE), Magdeburg, Germany. [31]Institute of Cognitive Neurology and Dementia Research (IKND), Otto-von-Guericke University, Magdeburg, Germany. [32]Center for Cognitive Disorders, Department of Psychiatry and Psychotherapy, Technical University of Munich, School of Medicine and Health, Klinikum rechts der Isar, Munich, Germany. [33]Department of Psychiatry and Psychotherapy, University Medical Center Goettingen, Goettingen, Germany. [34]German Center for Neurodegenerative Diseases (DZNE), Goettingen, Germany. [35]Medical Science Department, iBiMED, Aveiro, Portugal. [36]Institute of Human Genetics, University of Bonn, School of Medicine & University Hospital Bonn, Bonn, Germany. [37]Institute for Urban Public Health, University Hospital of University Duisburg-Essen, Essen, Germany. [38]1st Department of Neurology, Medical school, Aristotle University of Thessaloniki, Thessaloniki, Makedonia, Greece. [39]Taub Institute for Research in Alzheimer's Disease and the Aging Brain, The Gertrude H. Sergievsky Center, Depatment of Neurology, Columbia University, New York, NY, USA. [40]1st Department of Neurology, Aiginition Hospital, National and Kapodistrian University of Athens, Medical School, Athens, Greece. [41]Sant Pau Memory Unit, Institut de Recerca Sant Pau, Department of Neurology, Hospital de la Santa Creu i Sant Pau, Universitat Autònoma de Barcelona, Barcelona, Spain. [42]CIBERNED, Network Center for Biomedical Research in Neurodegenerative Diseases, National Institute of Health Carlos III, Madrid, Spain. [43]Department of Neurology. Hospital Universitario Donostia, San Sebastian, Spain. [44]Neurosciences Area, Instituto Biogipuzkoa, San Sebastian, Spain. [45]Unitat de Genètica Molecular, Institut de Biomedicina de València-CSIC, Valencia, Spain. [46]Centro de Biología Molecular Severo Ochoa (UAM-CSIC), Madrid, Spain. [47]Instituto de Investigacion Sanitaria 'Hospital la Paz' (IdIPaz), Madrid, Spain. [48]Universidad Autónoma de Madrid, Madrid, Spain. [49]Fundació Docència i Recerca MútuaTerrassa, Terrassa, Barcelona, Spain. [50]Memory Disorders Unit, Department of Neurology, Hospital Universitari Mutua de Terrassa, Terrassa, Barcelona, Spain. [51]Alzheimer's disease and other cognitive disorders unit. Service of Neurology. Hospital Clínic of Barcelona. Institut d'Investigacions Biomèdiques August Pi i Sunyer, University of Barcelona, Barcelona, Spain. [52]Laboratorio de Genética. Hospital Universitario Central de Asturias, Oviedo, Spain. [53]Instituto de Investigación Sanitaria del Principado de Asturias (ISPA), Asturias, Spain. [54]ACE Alzheimer Center Barcelona, Universitat Internacional de Catalunya, Barcelona, Spain. [55]Research Center and Memory clinic. ACE Alzheimer Center Barcelona, Universitat Internacional de Catalunya, Barcelona, Spain. [56]Unidad de Trastornos del Movimiento, Servicio de Neurología y Neurofisiología. Instituto de Biomedicina de Sevilla (IBiS), Hospital Universitario Virgen del Rocío/CSIC/Universidad de Sevilla, Seville, Spain. [57]Unidad Clínica de Enfermedades Infecciosas y Microbiología. Hospital Universitario de Valme, Sevilla, Spain. [58]Depatamento de Especialidades Quirúrgicas, Bioquímica e Inmunología. Facultad de Medicina. Universidad de Málaga, Málaga, Spain. [59]Unitat Trastorns Cognitius, Hospital Universitari Santa Maria de Lleida, Lleida, Spain. [60]Institut de Recerca Biomedica de Lleida (IRBLLeida), Lleida, Spain. [61]Alzheimer Research Center & Memory Clinic, Andalusian Institute for Neuroscience, Málaga, Spain. [62]Neurology Service, Marqués de Valdecilla University Hospital (University of Cantabria and IDIVAL), Santander, Spain. [63]Institute of Clinical Medicine - Neurology, University of Eastern Finland, Kuopio, Finland. [64]Institute of Biomedicine, University of Eastern Finland, Kuopio, Finland. [65]Faculty of Medicine, University of Lisbon, Lisbon, Portugal. [66]Clinic of Neurology, UH "Alexandrovska", Medical University - Sofia, Sofia, Bulgaria. [67]Department of Neurology, Medical University - Sofia, Sofia, Bulgaria. [68]Memory Clinic, Department of Neurology, Charles University, Second Faculty of Medicine and Motol University Hospital, Praha, Czech Republic. [69]Memory Clinic, Department of Neurology, Charles University, 2nd Faculty of Medicine and Motol University Hospital, Praha, Czech Republic. [70]International Clinical Research Center, St. Anne's University Hospital Brno, Brno, Czech Republic. [71]Department of Clinical Biochemistry, Copenhagen University Hospital - Rigshospitalet, Copenhagen, Denmark. [72]Department of Clinical Genetics, VU University Medical Centre, Amsterdam, The Netherlands. [73]Department of Neurology, ErasmusMC, Rotterdam, Netherlands. [74]Maastricht University, Department of Psychiatry & Neuropsychologie, Alzheimer Center Limburg, Maastricht, the Netherlands. [75]Unit for Hereditary Dementias, Theme Aging, Karolinska University Hospital-Solna, 171 64 Stockholm, Sweden. [76]Aging Research Center, Department of Neurobiology, Care Sciences and Society, Karolinska Institutet and Stockholm University, Stockholm, Sweden. [77]Dept.of Public Health and Carins Sciences / Geriatrics, Uppsala University, Uppsala, Sweden. [78]Université Paris-Saclay, CEA, Centre National de Recherche en Génomique Humaine, 91057 Evry, France. [79]Univ Rouen Normandie, Normandie Univ, Inserm U1245 and CHU Rouen, Department of Genetics and CNRMAJ, F-, 76000 Rouen, France. [80]Inserm, Bordeaux Population Health Research Center, UMR 1219, Univ. Bordeaux, ISPED, CIC 1401-EC, Univ Bordeaux, Bordeaux, France. [81]CHU de Bordeaux, Pole santé publique, Bordeaux, France. [82]Univ Lille Inserm 1171, CHU Clinical and Research Memory Research Centre (CMRR) of Distalz Lille France, Lille, France. [83]Université de Paris, EA 4468, APHP, Hôpital Broca, Paris, France. [84]University Bordeaux, Inserm, Bordeaux Population Health Research Center, Bordeaux, France. [85]Department of Neurology, Bordeaux University Hospital, Bordeaux, France. [86]Department of Child and Adolescent Psychiatry and Psychotherapy, University Hospital of Psychiatry Zurich, University of Zurich, Zurich, Switzerland. [87]Neuroscience Center Zurich, University of Zurich and ETH Zurich, Zurich, Switzerland. [88]Zurich Center for Integrative Human Physiology, University of Zurich, Zurich, Switzerland. [89]CHUV, Old Age Psychiatry, Department of Psychiatry, Lausanne, Switzerland. [90]Old Age Psychiatry, Department of Psychiatry, Lausanne University Hospital, Lausanne, Switzerland. [91]Department of Geriatric Psychiatry, University Hospital of Psychiatry Zürich, Zürich, Switzerland. [92]Molecular Markers Laboratory, IRCCS Istituto Centro San Giovanni di Dio Fatebenefratelli, Brescia, Italy. [93]Neurodegenerative Diseases Unit, Fondazione IRCCS Ca' Granda, Ospedale Policlinico, Milan, IT, Italy. [94]Dept. of Biomedical, Surgical and Dental Sciences, University of Milan, Milan, IT, Italy. [95]Department of Clinical Sciences and Community Health, University of Milan, 20122 Milan, Italy. [96]Geriatric Unit, Fondazione IRCCS Ca' Granda Ospedale Maggiore Policlinico, 20122 Milan, Italy. [97]Institute of Gerontology and Geriatrics, Department of Medicine and Surgery, University of Perugia, Perugia, Italy. [98]Division of Clinical Geriatrics, Department of Neurobiology, Care Sciences and

Society, Karolinska Institutet, Stockholm, Sweden. [99]Interdisciplinary Department of Medicine, Geriatric Medicine and Memory Unit, University of Bari "A. Moro", Bari, Italy. [100]Centre for Memory Disturbances, Lab of Clinical Neurochemistry, Section of Neurology, University of Perugia, Perugia, Italy. [101]Department of Biomedical Sciences, Section of Neuroscience and Clinical Pharmacology, University of Cagliari, Cagliari, Italy. [102]Neurology Unit, "San Gerardo" hospital, Monza and University of Milano-Bicocca, Milan, Italy. [103]Department of Clinical and Experimental Sciences, University of Brescia, Brescia, Italy. [104]Department of Cognitive Disorders and Old Age Psychiatry, University Hospital Bonn, Medical Faculty, Bonn, Germany. [105]Department of Neuroscience, Psychology, Drug Research and Child Health University of Florence, Florence, Italy. [106]IRCCS Fondazione Don Carlo Gnocchi, Florence, Italy. [107]Department of Medicine and Surgery, Unit of Neurology, University-Hospital of Parma, Parma, Italy. [108]Department of Hematology and Stem Cell Transplant, Vito Fazzi Hospital, Lecce, Italy. [109]Department of Neuroscience "Rita Levi Montalcini", University of Torino, Torino, Italy. [110]Department of Neuroscience, Università Cattolica del Sacro Cuore, Rome, Italy. [111]Neurology Unit, IRCCS Fondazione Policlinico Universitario A. Gemelli, Rome, Italy. [112]Laboratory of Neuropsychiatry, IRCCS Santa Lucia Foundation, Rome, Italy. [113]Institute of Neurology, Catholic University of the Sacred Heart, Rome, Italy. [114]Unit of Neurology V - Neuropathology, Fondazione IRCCS Istituto Neurologico Carlo Besta, Milan, Italy. [115]Department of Psychiatry and Psychotherapy, Faculty of Medicine and University Hospital Cologne, University of Cologne, Cologne, Germany. [116]Cluster of Excellence Cellular Stress Responses in Aging-associated Diseases (CECAD), University of Cologne, Cologne, Germany. [117]Translational Health Sciences, Bristol Medical School, University of Bristol, Bristol, UK. [118]Laboratory of Genetics, Immunology and Human Pathology, Faculty of Science of Tunis, University of Tunis El Manar, 2092 Tunis, Tunisia. [119]Alzheimer's Centre Reina Sofia-CIEN Foundation-ISCIII, 28031 Madrid, Spain. [120]Dept. of Public Health and Caring Sciences / Geriatrics, Uppsala University, Uppsala, Sweden. [121]Krembil Brain Institute, University Health Network, Toronto, Ontario, Canada. [122]Tanz Centre for Research in Neurodegenerative Diseases, Departments of Medicine and Laboratory Medicine & Pathobiology, University of Toronto, Toronto, Ontario, Canada. [123]NORMENT Centre, Division of Mental Health and Addiction, Oslo University Hospital, Oslo, Norway. [124]Institute of Clinical Medicine, University of Oslo, Oslo, Norway. [125]Research Center and Memory clinic Fundació ACE, Institut Català de Neurociències Aplicades, Universitat Internacional de Catalunya, Barcelona, Spain. [126]Biggs Institute for Alzheimer's and Neurodegenerative Diseases, University of Texas Health Science Center, San Antonio, Texas, USA. [127]Department of Psychiatry & Glenn Biggs Institute for Alzheimer's and Neurodegenerative Diseases, San Antonio, TX, USA. [128]Cologne Excellence Cluster on Cellular Stress Responses in Aging-Associated Disease (CECAD), University of Cologne, Cologne, Germany. [129]Department of Clinical Medicine, University of Copenhagen, Copenhagen, Denmark. [130]Department of Epidemiology, Erasmus MC, Rotterdam, The Netherlands. [131]Department of Radiology and Nuclear Medicine, Erasmus MC, Rotterdam, The Netherlands. [132]Department of Epidemiology, ErasmusMC, Rotterdam, The Netherlands. [267]These authors contributed equally: Matthew Bracher-Smith, Federico Melograna, Brittany Ulm. [268]These authors jointly supervised this work: Kristel Van Steen, Cornelia van Duijn, Valentina Escott-Price. *A list of authors and their affiliations appears at the end of the paper. ✉e-mail: kristel.vansteen@uliege.be; cornelia.vanduijn@ndph.ox.ac.uk; EscottPriceV@cardiff.ac.uk

## EADB

Iris Jansen[8,9], Sven van der Lee[8,9], Victor Andrade[133,134], Victoria Fernández[42,125], Maria-Carolina Dalmasso[135], Luca Kleineidam[135,136,137], Laura Molina-Porcel [28,29], Shahzad Ahmad[132,138], Dag Aarsland[139,140], Amanda Cano[42,125], Carla Abdelnour[42,125], Emilio Alarcón-Martín[125,141], Daniel Alcolea[41,42], Montserrat Alegret[42,125], Ignacio Alvarez[50,142], Nicola J. Armstrong[142], Tsolaki Anthoula[38,144], Ildebrando Appollonio[145,146], Marina Arcaro[147], Silvana Archetti[148], Alfonso Arias Pastor[59,60], Lavinia Athanasiu[149], Henri Bailly[83], Nerisa Banaj[150], Miquel Baquero[151], Ana Belén Pastor[152], Roberta Ghidoni [92], Claudine Berr[153], Céline Besse[78], Valentina Bessi[105,154], Giuliano Binetti[155], Silvia Fostinelli[155], Sonia Bellini[92], Alessandra Bizarro[156], Rafael Blesa[41,42], Mercè Boada[42,125], Silvia Boschi[109], Paola Bossù[157], Geir Bråthen[158,159], Catherine Bresner[160], Henry Brodaty[143,161], Keeley J. Brookes[162], Dolores Buiza-Rueda[42,56], Katharina Bûrger[163,164], Vanessa Burholt[165,166], Miguel Calero[42,152,167], Geneviève Chene[80,81], Ángel Carracedo[168,169], Roberta Cecchetti[97], Laura Cervera-Carles[41,42], Camille Charbonnier[170], Caterina Chillotti[171], Simona Ciccone[172], Jurgen A. H. R. Claassen[173], Jordi Clarimon[41,42], Christopher Clark[174], Elisa Conti[145], Anaïs Corma-Gómez[57], Antonio Daniele[110,111], Guido Maria Giuffrè[110], Carlo Custodero[175], Delphine Daian[78], Efthimios Dardiotis[176], Jean-François Dartigues[84], Peter Paul de Deyn[177], Teodoro del Ser[178], Nicola Denning[1], Janine Diehl-Schmid[179], Mónica Diez-Fairen[50,142], Paolo Dionigi Rossi[172], Srdjan Djurovic[149], Emmanuelle Duron[83], Sebastiaan Engelborghs[180,181,182,183], Josep Blázquez[42,125], Michael Ewers[163,164], Tagliavini Fabrizio[184], Sune Fallgaard Nielsen[185], Lucia Farotti[100], Chiara Fenoglio[186], Marta Fernández-Fuertes[57], Catarina B. Ferreira[65], Evelyn Ferri[172], Bertrand Fin[78], Peter Fischer[187], Tormod Fladby[124], Klaus Fließbach[137,188], Juan Fortea[41,42], Tatiana M. Foroud[189], Silvia Fostinelli[92], Nick C. Fox[190], Emlio Franco-Macías[191], Ana Frank-García[42,47,192], Lutz Froelich[193], Jose Maria García-Alberca[42,61], Pablo García-González[125], Sebastian Garcia-Madrona[194], Guillermo Garcia-Ribas[194], Ina Giegling[195], Giaccone Giorgio[184], Oliver Goldhardt[179], Antonio González-Pérez[196], Giulia Grande[76], Emma Green[197], Edna Grünblatt [86,87,88], Tamar Guetta-Baranes[198], Annakaisa Haapasalo[199], Georgios Hadjigeorgiou[200], Harald Hampel[201,202], John Hardy[203], Annette M. Hartmann[195], Ganna Leonenko[160], Janet Harwood[160], Seppo Helisalmi[204,205], Michael T. Heneka[206,207], Isabel Hernández[42,125], Martin J. Herrmann[208], Per Hoffmann[36], Clive Holmes[209], Raquel Huerto Vilas[59,60], Marc Hulsman[8,210], Geert Jan Biessels[211], Charlotte Johansson[212,213], Lena Kilander[77], Anne Kinhult Ståhlbom[212,213], Miia Kivipelto[214,215,216,217], Anne Koivisto[204,218,219], Johannes Kornhuber[220], Mary H. Kosmidis[221], Carmen Lage[42,62], Erika J. Laukka[76,222], Alessandra Lauria[156], Jenni Lehtisalo[204,223], Ondrej Lerch[69,70], Alberto Lleó[41,42], Adolfo Lopez de Munain[42,224], Seth Love[225], Malin Löwemark[77], Lauren Luckcuck[160], Juan Macías[57], Catherine A. MacLeod[226], Wolfgang Maier[134,137], Francesca Mangialasche[214], Spallazzi Marco[227], Marta Marquié[42,125], Rachel Marshall[160], Angel Martín Montes[42,47,192],

Carmen Martínez Rodríguez[228], Simon Mead[229], Miguel Medina[42,152], Alun Meggy[1], Silvia Mendoza[61], Manuel Menéndez-González[228], Pablo Mir [42,56], Merel Mol[73], Laura Montrreal[125], Kevin Morgan[230], Markus M. Nöthen[36], Tiia Ngandu[223], Børge G. Nordestgaard[129,231], Robert Olaso[78], Adelina Orellana[42,125], Michela Orsini[232], Maria Capdevila[42,125], Alessandro Padovani[233], Caffarra Paolo[234], Marta Martinez-Lucas[125], Pierre Pericard[235], Juan A. Pineda[57], Gerard Piñol-Ripoll[59,60], Claudia Pisanu[236], Thomas Polak[208], Danielle Posthuma[9], Josef Priller[16,237], Raquel Puerta[125], Olivier Quenez[79], Inés Quintela[168], Alberto Rábano[42,152,238], Luis M. Real [57,58], Marcel J. T. Reinders[239], Peter Riederer[240], Claudia Olivé[125], Eloy Rodriguez-Rodriguez [42,62], Arvid Rongve[241,242], Irene Rosas Allende[52,53], Maitée Rosende-Roca[42,125], Jose Luis Royo[243], Elisa Rubino[244], María Eugenia Sáez[196], Paraskevi Sakka[245], Ingvild Saltvedt[159,246], Fernando García-Gutierrez[42,125], María Bernal Sánchez-Arjona[191], Florentino Sanchez-Garcia[247], Pascual Sánchez-Juan[42,62], Raquel Sánchez-Valle[248], Sigrid B. Sando[158,159], Michela Scamosci[97], Elio Scarpini[147,186], Martin Scherer[249], Matthias Schmid[207,250], Jonathan M. Schott[190], Geir Selbæk[124,251], Alexey A. Shadrin[149], Olivia Skrobot[225], Alina Solomon[204,214], Sandro Sorbi[105,106], Oscar Sotolongo-Grau[125], Annika Spottke[207,252], Eystein Stordal[253], Andrea Miguel[125], Lluís Tárraga[42,125], Niccolo Tesi[8,210], Anbupalam Thalamuthu[143], Tegos Thomas[38], Latchezar Traykov[66], Anne Tybjærg-Hansen[71,129], Andre Uitterlinden[254], Abbe Ullgren[212], Ingun Ulstein[251], Sergi Valero[42,125], Christine Van Broeckhoven[255,256,257], Jasper Van Dongen[14,255,256], Jeroen van Rooij[73,258], Rik Vandenberghe[259,260], Jean-Sebastian Vidal[83], Maria Gabriella Vita[110], Jonathan Vogelgsang[33,261], Michael Wagner[188,207], David Wallon[262], Leonie Weinhold[250], Gill Windle[226], Bob Woods[226], Mary Yannakoulia[263], Miren Zulaica[42,264], Mohsen Ghanbari[132], Perminder Sachdev[143,265], Karen Mather[143] & Mohammad Arfan Ikram[266]

[133]Division of Neurogenetics and Molecular Psychiatry, Department of Psychiatry and Psychotherapy, University of Cologne, Medical Faculty, Cologne, Germany. [134]Department of Neurodegenerative Diseases and Geriatric Psychiatry, University Hospital Bonn, Bonn, Germany. [135]Division of Neurogenetics and Molecular Psychiatry, Department of Psychiatry and Psychotherapy, University of Cologne, Medical Faculty, 50937 Cologne, Germany. [136]Department of Cognitive Disorders and Old Age Psychiatry, University of Bonn, Bonn, Germany. [137]German Center for Neurodegenerative Diseases (DZNE Bonn), Bonn, Germany. [138]LACDR, Leiden, The Netherlands. [139]Centre of Age-Related Medicine, Stavanger University Hospital, Stavanger, Norway. [140]Institute of Psychiatry, Psychology & Neuroscience, PO 70, 16 De Crespigny Park, London SE58AF, UK. [141]Department of Surgery, Biochemistry and Molecular Biology, School of Medicine, University of Málaga, Málaga, Spain. [142]Fundació Docència i Recerca MútuaTerrassa and Movement Disorders Unit, Department of Neurology, University Hospital MútuaTerrassa, Terrassa, 08221 Barcelona, Spain. [143]Centre for Healthy Brain Ageing, School of Psychiatry, Faculty of Medicine, University of New South Wales, Sydney, Australia. [144]Alzheimer Hellas, Thessaloniki, Makedonia, Greece. [145]School of Medicine and Surgery, University of Milano-Bicocca, Milano, Italy. [146]Neurology Unit, "San Gerardo" hospital, Monza, Italy. [147]Fondazione IRCCS Ca' Granda, Ospedale Policlinico, Milan, Italy. [148]Department of Laboratory Diagnostics, III Laboratory of Analysis, Brescia Hospital, Brescia, Italy. [149]NORMENT Centre, University of Oslo, Oslo, Norway. [150]Laboratory of Neuropsychiatry, Department of Clinical and Behavioral Neurology, IRCCS Santa Lucia Foundation, Rome, Italy. [151]Servei de Neurologia, Hospital Universitari i Politècnic La Fe, Valencia, Spain. [152]CIEN Foundation/Queen Sofia Foundation Alzheimer Center, Madrid, Spain. [153]Univ. Montpellier, Inserm U1061, Neuropsychiatry: epidemiological and clinical research, PSNREC, Montpellier, France. [154]Azienda Ospedaliero-Universitaria Careggi, Florence, Italy. [155]MAC-Memory Clinic and Molecular Markers Laboratory, IRCCS Istituto Centro San Giovanni di Dio Fatebenefratelli, Brescia, Italy. [156]Geriatrics Unit Fondazione Policlinico A. Gemelli IRCCS, Rome, Italy. [157]Experimental Neuro-psychobiology Laboratory,Department of Clinical and Behavioral Neurology, IRCCS Santa Lucia Foundation, Rome, Italy. [158]Department of Neurology and Clinical Neurophysiology, University Hospital of Trondheim, Trondheim, Norway. [159]Department of Neuromedicine and Movement Science, Norwegian University of Science and Technology, Trondheim, Norway. [160]MRC Centre for Neuropsychiatric Genetics and Genomics, Division of Psychological Medicine and Clinical Neuroscience, School of Medicine, Cardiff University, Cardiff, UK. [161]Dementia Centre for Research Collaboration, School of Psychiatry, University of New South Wales, Sydney, Australia. [162]Biosciences, School of Science and Technology, Nottingham Trent University, Nottingham, UK. [163]Institute for Stroke and Dementia Research, Klinikum der Universität München, Ludwig-Maximilians-Universität LMU, Munich, Germany. [164]German Center for Neurodegenerative Diseases (DZNE, Munich), Munich, Germany. [165]Faculty of Medical & Health Sciences, University of Auckland, Auckland, New Zealand. [166]Wales Centre for Ageing & Dementia Research, Swansea University, Wales, New Zealand. [167]UFIEC, Instituto de Salud Carlos III, Madrid, Spain. [168]Grupo de Medicina Xenómica, Centro Nacional de Genotipado (CEGEN-PRB3-ISCIII). Universidade de Santiago de Compostela, Santiago de Compostela, Spain. [169]Fundación Pública Galega de Medicina Xenómica- CIBERER-IDIS, University of Santiago de Compostela, Santiago de Compostela, Spain. [170]Univ Rouen Normandie, Normandie Univ, Inserm U1245 and CHU Rouen, Department of Biostatistics and CNRMAJ, F-76000 Rouen, France. [171]Unit of Clinical Pharmacology, University Hospital of Cagliari, Cagliari, Italy. [172]Geriatic Unit, Fondazione Cà Granda, IRCCS Ospedale Maggiore Policlinico, Milan, Italy. [173]Radboudumc Alzheimer Center, Department of Geriatrics, Radboud University Medical Center, Nijmegen, the Netherlands. [174]Institute for Regenerative Medicine, University of Zürich, Schlieren, Switzerland. [175]University of Bari, "A. Moro", Bari, Italy. [176]School of Medicine, University of Thessaly, Larissa, Greece. [177]Department of Neurology, University Medical Center Groningen, Groningen, the Netherlands. [178]Department of Neurology/CIEN Foundation/Queen Sofia Foundation Alzheimer Center, Madrid, Spain. [179]Technical University of Munich, School of Medicine, Klinikum rechts der Isar, Department of Psychiatry and Psychotherapy, Munich, Germany. [180]Center for Neurosciences, Vrije Universiteit Brussel (VUB), Brussels, Belgium. [181]Reference Center for Biological Markers of Dementia (BIODEM), Institute Born-Bunge, University of Antwerp, Antwerp, Belgium. [182]Institute Born-Bunge, University of Antwerp, Antwerp, Belgium. [183]Department of Neurology, UZ Brussel, Brussels, Belgium. [184]Fondazione IRCCS, Istituto Neurologico Carlo Besta, Milan, Italy. [185]Department of Clinical Biochemistry, Herlev and Gentofte Hospital, Herlev, Denmark. [186]University of Milan, Milan, Italy. [187]Department of Psychiatry, Social Medicine Center East- Donauspital, Vienna, Austria. [188]Department of Neurodegeneration and Geriatric Psychiatry, University of Bonn, 53127 Bonn, Germany. [189]Department of Medical and Molecular Genetics, Indiana University, Indianapolis, Indiana, USA. [190]Dementia Research Centre, UCL Queen Square Institute of Neurology, London, United Kingdom. [191]Unidad de Demencias, Servicio de Neurología y Neurofisiología. Instituto de Biomedicina de Sevilla (IBiS), Hospital Universitario Virgen del Rocío/CSIC/Universidad de Sevilla, Seville, Spain. [192]Hospital Universitario la Paz, Madrid, Spain. [193]Department of geriatric Psychiatry, Central Institute for Mental Health, Mannheim, University of Heidelberg, Mannheim, Germany. [194]Hospital Universitario Ramon y Cajal, IRYCIS, Madrid, Spain. [195]Department of Psychiatry and Psychotherapy, Medical University of Vienna, Vienna, Austria. [196]CAEBI, Centro Andaluz de Estudios Bioinformáticos, Sevilla, Spain. [197]Institute of Public

Health, University of Cambridge, Cambridge, UK. [198]Human Genetics, School of Life Sciences, Life Sciences Building, University Park, University of Nottingham, Nottingham, UK. [199]A.I Virtanen Institute for Molecular Sciences, University of Eastern Finland, Kuopio, Finland. [200]Department of Neurology, Medical School, University of Cyprus, Nicosia, Cyprus. [201]Sorbonne University, GRC n° 21, Alzheimer Precision Medicine Initiative (APMI), AP-HP, Pitié-Salpêtrière Hospital, Boulevard de l'hôpital, F-, 75013 Paris, France. [202]Eisai Inc., Neurology Business Group, 100 Tice Blvd, Woodcliff Lake, NJ 07677, USA. [203]Reta Lila Weston Research Laboratories, Department of Molecular Neuroscience, UCL Institute of Neurology, London, UK. [204]Insitute of Clinical Medicine - Neurology, University of Eastern, Kuopio, Finland. [205]Institute of Clinical Medicine – Internal Medicine, University of Eastern Finland, Kuopio, Finland. [206]Department of Neurodegeneration and Geriatric Psychiatry, University of Bonn, Bonn, Germany. [207]German Center for Neurodegenerative Diseases (DZNE, Bonn), Bonn, Germany. [208]Department of Psychiatry, Psychosomatics and Psychotherapy, Center of Mental Health, University Hospital, Wuerzburg, Germany. [209]Clinical and Experimental Science, Faculty of Medicine, University of Southampton, Southampton, UK. [210]Section Genomics of Neurdegenerative Diseases and Aging, Department of Human Genetics Amsterdam UMC, Vrije Universiteit Amsterdam, Amsterdam UMC, Amsterdam, The Netherlands. [211]Department of Neurology, UMC Utrecht Brain Center, Utrecht, the Netherlands. [212]Karolinska Institutet, Center for Alzheimer Research, Department NVS, Division of Neurogeriatrics, Stockholm, Sweden. [213]Unit for Hereditary dementias, Karolinska University Hospital-Solna, Stockholm, Sweden. [214]Division of Clinical Geriatrics, Center for Alzheimer Research, Care Sciences and Society (NVS), Karolinska Institutet, Stockholm, Sweden. [215]Institute of Public Health and Clinical Nutrition, University of Eastern Finland, Kuopio, Finland. [216]Neuroepidemiology and Ageing Research Unit, School of Public Health, Imperial College London, London, United Kingdom. [217]Stockholms Sjukhem, Research & Development Unit, Stockholm, Sweden. [218]Department of Neurology, Kuopio University Hospital, Kuopio, Finland. [219]Department of Neurosciences, University of Helsinki and Department of Geriatrics, Helsinki University Hospital, Helsinki, Finland. [220]Department of Psychiatry and Psychotherapy, Universitätsklinikum Erlangen, and Friedrich-Alexander Universität Erlangen-Nürnberg, Erlangen, Germany. [221]Laboratory of Cognitive Neuroscience, School of Psychology, Aristotle University of Thessaloniki, Thessaloniki, Greece. [222]Stockholm Gerontology Research Center, Stockholm, Sweden. [223]Public Health Promotion Unit, Finnish Institute for Health and Welfare, Helsinki, Finland. [224]Department of Neurology. Hospital Universitario Donostia. OSAKIDETZA-Servicio Vasco de Salud, San Sebastian, Spain. [225]Translational Health Sciences, Bristol Medical School, University of Bristol, Bristol BS16 1LE, UK. [226]School of Health Sciences, Bangor University, Bangor, UK. [227]Unit of Neurology, University of Parma and AOU, Parma, Italy. [228]Servicio de Neurología HOspital Universitario Central de Asturias- Oviedo and Instituto de Investigación Biosanitaria del Principado de Asturias, Oviedo, Spain. [229]MRC Prion Unit at UCL, UCL Institute of Prion Diseases, London W1W 7FF, UK. [230]Human Genetics, School of Life Sciences, University of Nottingham, NG7 2UH Nottingham, UK. [231]Department of Clinical Biochemistry, Copenhagen University Hospital - Herlev and Gentofte, Herlev, Denmark. [232]Department of Neuroscience, Catholic University of Sacred Heart, Fondazione Policlinico Universitario A. Gemelli IRCCS, Rome, Italy. [233]Centre for Neurodegenerative Disorders, Department of Clinical and Experimental Sciences, University of Brescia, Brescia, Italy. [234]DIMEC, University of Parma, Parma, Italy. [235]Univ. Lille, CNRS, Inserm, CHU Lille, Institut Pasteur de Lille, US 41-UMS 2014-PLBS, bilille, Lille, France. [236]Department of Biomedical Sciences, University of Cagliari, Cagliari, Italy. [237]Department of Neuropsychiatry and Laboratory of Molecular Psychiatry, Charité, Charitéplatz 1, 10117 Berlin, Germany. [238]BT-CIEN, Madrid, Spain. [239]Delft Bioinformatics Lab, Delft University of Technology, Delft, The Netherlands. [240]Center of Mental Health, Clinic and Policlinic of Psychiatry, Psychosomatics and Psychotherapy, University Hospital of Würzburg, Wuerzburg, Germany. [241]Department of Research and Innovation, Helse Fonna, Haugesund Hospital, Haugesund, Norway. [242]The University of Bergen, Institute of Clinical Medicine (K1), Bergen, Norway. [243]Departamento de Especialidades Quirúrgicas, Bioquímicas e Inmunología, School of Medicine, University of Málaga, Málaga, Spain. [244]Department of Neuroscience and Mental Health, AOU Città della Salute e della Scienza di Torino, Torino, Italy. [245]Athens Association of Alzheimer's disease and Related Disorders, Athens, Greece. [246]Department of Geriatrics, St. Olav's Hospital, Trondheim University Hospital, Trondheim, Norway. [247]Department of Immunology, Hospital Universitario Doctor Negrín, Las Palmas de Gran Canaria, Las Palmas, Spain. [248]Neurology department-Hospital Clínic, IDIBAPS, Universitat de Barcelona, Barcelona, Spain. [249]Department of Primary Medical Care, University Medical Centre Hamburg-Eppendorf, 20246 Hamburg, Germany. [250]Institute of Medical Biometry, Informatics and Epidemiology, University Hospital of Bonn, Bonn, Germany. [251]Department of Geriatric Medicine, Oslo University Hospital, Oslo, Norway. [252]Department of Neurology, University of Bonn, Bonn, Germany. [253]Department of Psychiatry, Namsos Hospital, Namsos, Norway. [254]Department of Internal medicine and Biostatistics, ErasmusMC, Rotterdam, The Netherlands. [255]Laboratory of Neurogenetics, Institute Born - Bunge, Antwerp, Belgium. [256]Department of Biomedical Sciences, University of Antwerp, Neurodegenerative Brain Diseases Group, Center for Molecular Neurology, VIB, Antwerp, Belgium. [257]Neurodegenerative Brain Diseases Group, VIB Center for Molecular Neurology, VIB, Antwerp, Belgium. [258]Department of Internal Medicine, ErasmusMC, Rotterdam, The Netherlands. [259]Laboratory for Cognitive Neurology, Department of Neurosciences, University of Leuven, Leuven, Belgium. [260]Neurology Department, University Hospitals Leuven, Leuven, Belgium. [261]Department of Psychiatry, Harvard Medical School, McLean Hospital, Belmont, MA, USA. [262]Univ Rouen Normandie, Normandie Univ, Inserm U1245 and CHU Rouen, Department of Neurology and CNRMAJ, F-76000 Rouen, France. [263]Department of Nutrition and Diatetics, Harokopio University, Athens, Greece. [264]Neurosciences Area. Instituto Biodonostia, San Sebastian, Spain. [265]Neuropsychiatric Institute, Prince of Wales Hospital, Sydney, Australia. [266]Department of Epidemiology, Erasmus University Medical Center, Rotterdam, Netherlands.

