## [Peer Review file · Nature Communications]

Machine learning in Alzheimer's disease genetics

Corresponding Author: Professor Valentina Escott-Price

Version 0:

Reviewer comments:

Reviewer #1

(Remarks to the Author)

This paper performed machine learning analysis on the genotyping data from the European Alzheimer & Dementia Biobank (EADB) to identify AD related genetic determinants. The authors examined three machine learning methods: 1) Gradient Boosting Machines (GBM), 2) biological pathway-informed Neural Networks (NN), and 3) Model-based Multifactor Dimensionality Reduction (MBMDR). They compared the prediction results of these three methods with a naïve polygenic risk score (PRS) based method. The AUC ranged from 0.670 (MBMDR) to 0.692 (GMB). Model predictions showed pairwise correlations ranging from 0.75 to 0.86 (Pearson's r) and from 0.80 to 0.87 (Spearman's r). Functional annotation (e.g., enrichment analysis) was performed on the identified SNPs. The studied problem is interesting. The paper is well-structured and easy to follow. The methods are well-established ones. The results are interesting. The following are a few comments that I hope can further improve the manuscript.

- The authors selected three ML methods, including GBM, NN and MBMDR. There are many ML methods available. It would be helpful to provide some discussion on why these methods were selected in this work.
- There is a lack of replication study. The authors only examined one single training-testing partition, Models were trained using training data and then applied to the testing data. The testing performances were reported. Feature importance was evaluated using model specific methods to identify top features. There is only one set of results reported. It is unclear whether it is specific to that specific partition. It would be helpful to examine multiple random partitions and check whether the results (both AUC and SNP identification) are stable.
- The genetic data is very high dimensional (>200K SNPs). The number of SNPs is much larger than the number of training samples (29K). There is a high risk for overfitting. It would be helpful to provide some relevant discussion. In addition, the data is large. It would also be help to provide some discussion on running time.
- "For ML based analysis, we limited the data to genotyped (non-imputed) variants. We additionally integrated imputed lead independent SNPs previously reported as genome wide significant 21,23,24 after converting to best guess genotypes (threshold = 0.9); quality control was applied as described previously 21." This procedure has a potential risk of circularity. If any of EADB samples were included in the prior studies to genetic those significant SNPs, the procedure is biased. The testing data would become no longer independent, since they were used to derive those significant SNPs. Any relevant discussion on this topic would be helpful.
- "To give the same information to both PRS and all machine learning models, PRS were derived using summary statistics which were generated in the training set by running a genome-wide association study (GWAS), adjusting for the same confounders in PLINK 320 v2.00a3.3LM 37 (Supplementary section 1.5)". It is unclear what p-value threshold was used to derive the PRS. Difference thresholds would lead to different prediction performances.
- "The top SNPs identified by machine learning were selected by the permutation-based p-value threshold defined above for MB-MDR ($p_{adj} < 1$), empirically by taking the extreme tail of the distribution for SHAP values ($\mu|SHAP| > 0.0005$) in gradient boosting (Supplementary Figure 3), summed weights ($p_{adj} < 0.05$) for neural networks (Supplementary Figure 5), and by applying the Boruta algorithm 38 (gradient boosting models only). Details on predictor selection are given in Supplementary sections 1.6 and 2.2-2.4." Different methods for determine relevant features were used among three ML models. It is unclear whether these SNP identification methods are comparable among one another.

• “Known lead variants in APOE and BIN1 were the most important predictors in GBM and NNs, with causal SNPs for deriving APOE status identified and ranked as the most important SNPs by both GBMs and NNs (Figure 3).” The meaning of “causal SNPs” is unclear. It would be helpful to provide further clarification.

In summary, the value of the results presented here is unclear, given that they were not replicated yet or their stability was not examined yet. The significance of the work is unclear either, given that the criteria for generating ML-derived SNPs need further justification. The current work does not support the conclusions and claims. The additional evidence to address the above concerns is needed.

(Remarks on code availability)

The github repositories were well-documented. I did not test the code.

Reviewer #2

(Remarks to the Author)

Cool paper + big and solid team.

Some comments:

- "co-dominant" in reality should be dosage or additive.
- Gridsearch run details or bayes opt params for hyper tune?
- "multiplicative relationship between SNPs" needs more detail regarding allele direction, 0 count compensation (+ 1) and if this is main (adjusting for SNP effects themselves separately) or joint effects. Additionally, some of the algorithms used will be including interactions or tree based proxies with information content representing interactions.
- "IDUA and DGKQ" also other diseases of the brain like Parkinson's, IIRC.
- Please provide more detailed performance stats such as balanced accuracy, sensitivity, specificity as well as prevalence adjusted TPR + NPR etc.
- Why those algorithms for testing?
- More details are needed on accounting for covariates. Some people go as far as adjusting genotypes for the covariates then running the training after normalization. How was numeric space normalized for covariates in models, Z scale?
- Static 70:30 for all algorithms, same samples in each algo test?

Good luck!

Mike Nalls

(Remarks on code availability)

Reviewer #3

(Remarks to the Author)

This is an interesting and informative study in terms of topic and comparing machine learning and statistical methods for analysing genomic data.

The study uses data from the European Alzheimer & Dementia Biobank (EADB) consortium, which combines genetic and clinical data from 15 countries. All participants included in the study were unrelated individuals of European ancestry, comprising 20,013 clinically defined AD cases and 21,673 controls, after excluding participants present in Kunkle et al.

The study has demonstrated the similarities and differences between ML and traditional genetic risk modelling methods as well as several novel genes/pathways identified by ML but not by traditional modelling –this is especially interesting.

To build the machine learning models, the authors split the participants were randomly separated into a 70%-30% train-test split, resulting in 29,180 individuals in the training set (14,006 cases; 15,174 controls) and 12,506 for testing (6,007 cases; 6,499 controls), each with 215,193 predictors after quality control procedures.

The pre-processing and machine learning pipelines are clearly described. The supplementary material also provides more detailed information about the correlation analysis and performance of the models.

The scale of the data used in the study offers statistical power, allowing for reliable assessment of potential confounders. The quality control methods are robust. The use of biologically informed neural network is a good example of how machine learning models can mimic existing information structures. It is also especially interesting to read about the novel SNPs, genes, and pathways that the ML models identified.

The findings are evaluated based on the identified loci and SNPs via the models, and these findings are compared with known literature.

While the results and the proposed models are interesting, it would be helpful if the authors could further clarify how these findings add to the understanding of AD genomics and the current literature in this area.

Regarding the method and results:

- The results of the three machine learning models are compared using a statistical test, and it is reported that they all performed similarly on the test data (AUC=0.69). This statistical test provides interesting insight into the fact that the models predicted the outcomes in the same direction. However, it would be very informative if the authors could report the prediction task results using specificity sensitivity metrics. This would have helped to better understand the performance of the models.

- In relation to the above, the AUC is not high, and especially without including the APOE region, the AUC is around 0.59; the confidence in findings and feature importance extracted from the model to derive the statical findings should also be treated with caution. It would have been useful if further improvements on the models could be made to improve the prediction of the model, and then the feature analysis would have been performed on the best performing model. Showing that the three models (with similar predictive power and similar input) perform similarly does not improve the confidence in the results.

- The presented AUC does not report confidence intervals or variations across different runs. While a 70/30 split is used for training/test and 4-fold cross-validation is applied for hyperparameter tuning, a bootstrapping method could be applied to the training set to demonstrate variations of the metrics across multiple runs.

- In the experiment excluding APOE, it would be useful if the authors had identified APOE carriers of the participants with known AD status and then ran an experiment to show how the prediction results could vary by excluding the APOE SNPs for APOE carrier/non-carriers.

- The authors have indicated that the previous studies using ML had the risk of bias. It would have been helpful if the authors had conducted a post-model prediction analysis to report on the differences in predictions (if any) according to sex, age, and other demographic data.

- The authors mention that (line 605) "Finally, selection of genes for interpretation of models' results was somewhat arbitrary, as there are no standard thresholds for important feature selection in different ML approaches; this may partially explain the incomplete overlap between top SNPs identified between different ML approaches." Would it be possible to choose for example top n features based on the SHAP values and then perform a survey analysis to say out of the "n" top features how many of them were previously known and reported in literature?

- I'm curious why you think different ML models identified SNPs that were enriched in different cell types. You wrote: "SNPs prioritised by GBM were significantly enriched in microglial regions and showed greater enrichment in astrocytes when modelled without the APOE region. Findings from NN and MBMDR were enriched in synaptic regions." Please explain why you think this is in more detail.

- In your Quality Control section, you write: "For ML based analysis, we limited the data to genotyped (non-imputed) variants. We additionally integrated imputed lead independent SNPs previously reported as genome-wide significant after converting to best guess genotypes" – these sentences seem contradictory; did you use imputed values or not?

- Please ensure all abbreviations are appropriately defined. Ex) MAF is not listed in parentheses after "minor allele frequency".

- The authors have assessed the power of ML by measuring the correlation between ML and traditional models – does this necessarily measure the quality of ML as a novel approach to modelling disease risk, or simply how similar it is to existing models? As far as I understand, the purpose of this work was to determine how ML can surpass traditional methods in modelling disease, couldn't that come at a cost of its correlation with existing methods?

- The use of federated learning models is mentioned as one of the future directions. The privacy requirement is clear, but in cases where the data can be combined (such as this example), there are other ways to improve the homogeneity of the data across multiple sites/sources. It would have been helpful if some of these approaches would have been mentioned.

(Remarks on code availability)

I did not review the code, but the GitHub repositories are well-structured and the DAXOS: Dask and XGBoost on SNPs includes the installation, usage and test instructions. This will help with the reproducibility of the work.

Reviewer #4

(Remarks to the Author)

(Remarks on code availability)

N/A

Version 1:

Reviewer comments:

Reviewer #1

(Remarks to the Author)

My comments have been properly addressed.

(Remarks on code availability)

Reviewer #2

(Remarks to the Author)

Thanks for the revisions.

- Still not sure if 0 count compensation on the formal interaction analysis was used when the regression was run. Without this, dynamic range is flattened for all major hmozygotes.

- Please be clear that you are using probabilistic AUCs not including specific cut off values.

- Code and data availability could be a little clearer, but we all understand painful logistics.

Good luck!

M.A. Nalls

(Remarks on code availability)

These are packages but not the minutae of the paper itself. Generally seems fine.

Reviewer #3

(Remarks to the Author)

I would like to thank the authors for revising the paper and for responding to the review comments. The authors have responded to most of our comments and concerns.

Only two minor suggestions:

1. The sensitivity and specificity are relatively low, and the authors also agree that running an experiment to show how the prediction results could vary by excluding the APOE SNPs for APOE carrier/non-carriers would be helpful. The authors' response is added below.

"We agree with the reviewer and considered running these analyses as part of the original manuscript. However, we believe that fully addressing this question requires a separate study design where analyses are run separately in all APOE strata, especially e44, e4 and e33 carriers separately. The consortium (EADB), which has kindly provided us these data, has a separate e4-stratification analysis group, which is looking at the APOE-stratified analyses."

In this case, please add a subsection to the discussion section and clearly state this as a limitation to their current study and a potential area for future work.

2. The differences based on the female/male split are shown in Figure 2 (i); the panel is difficult to see/interpret; it would be good if the authors also included one or two sentences about the results of this new experiment in the paper.

(Remarks on code availability)

n/a

REVIEWER COMMENTS

Reviewer #1 (Remarks to the Author):

This paper performed machine learning analysis on the genotyping data from the European Alzheimer & Dementia Biobank (EADB) to identify AD related genetic determinants. The authors examined three machine learning methods: 1) Gradient Boosting Machines (GBM), 2) biological pathway-informed Neural Networks (NN), and 3) Model-based Multifactor Dimensionality Reduction (MBMDR). They compared the prediction results of these three methods with a naïve polygenic risk score (PRS) based method. The AUC ranged from 0.670 (MBMDR) to 0.692 (GMB). Model predictions showed pairwise correlations ranging from 0.75 to 0.86 (Pearson's r) and from 0.80 to 0.87 (Spearman's r). Functional annotation (e.g., enrichment analysis) was performed on the identified SNPs. The studied problem is interesting. The paper is well-structured and easy to follow. The methods are well-established ones. The results are interesting. The following are a few comments that I hope can further improve the manuscript.

We thank the reviewer for their overview and helpful comments. We have revised the manuscript and believe it is significantly stronger for it. We respond to each point in kind below.

- The authors selected three ML methods, including GBM, NN and MBMDR. There are many ML methods available. It would be helpful to provide some discussion on why these methods were selected in this work.

We focus on these three methods due to their prominence in the field, demonstrated performance, and complementary strengths. GBMs perform particularly well across a wide range of data and are the most frequently used model as it is easy to implement and usually quick to run. Neural networks are highly flexible and have the ability to easily embed prior knowledge in their architecture, as implemented here, while MB-MDR is particularly well-suited to detecting SNP-SNP interactions (<https://doi.org/10.1093/bib/bbr012>). We have included information to this effect in the discussion (page 16, lines 467-472).

- There is a lack of replication study. The authors only examined one single training-testing partition. Models were trained using training data and then applied to the testing data. The testing performances were reported. Feature importance was evaluated using model specific methods to identify top features. There is only one set of results reported. It is unclear whether it is specific to that specific partition. It would be helpful to examine multiple random partitions and check whether the results (both AUC and SNP identification) are stable.

We thank the reviewer for this point. We have now performed both replication in external data and stability tests on the models within the training data. We performed repeated runs of all ML models with different random train-test splits in order to examine the stability of the models. We note that the performance of these models is consistent across random splits (page 13, lines 389-390; Supplementary Table 4). We only take forward stable SNPs which are present in at least two repeats for a given model, increasing the likelihood that results are robust. Furthermore, we test for replication in an external dataset for these SNPs (methods: page 11, lines 338-349), highlighting 6 novel loci which pass these more stringent criteria for associated to AD (page 14, lines 400-414), which we now highlight in the

discussion (page 18, 498-526). Updated loci can also be seen in Table 1 alongside replication *p*-values.

- The genetic data is very high dimensional (>200K SNPs). The number of SNPs is much larger than the number of training samples (29K). There is a high risk for overfitting. It would be helpful to provide some relevant discussion. In addition, the data is large. It would also be help to provide some discussion on running time.

We agree with the reviewer that dimensions are particularly high in relation to number of observations, increasing the risk of overfitting. To mitigate this risk, we applied penalisation to models where possible, with hyperparameters derived from cross-validation, ensuring they maximise test AUC. We have now highlighted this point in the methods (page 9, lines 282-287). Additionally, training set AUC for the best performing model (GBMs) is only 0.8 before covariate adjustment. As this is from predicting on the exact data used for training, a much higher AUC would be expected if overfitting were present. We also note that all performance estimates are calculated on test data which was unseen in training (page 8, lines 271-274). These do not show a marked increase over standard linear models (PRS) (page 13, lines 384-387; Supplementary table 3) so inflation of results through overfitting is unlikely here.

- “For ML based analysis, we limited the data to genotyped (non-imputed) variants. We additionally integrated imputed lead independent SNPs previously reported as genome wide significant 21,23,24 after converting to best guess genotypes (threshold = 0.9); quality control was applied as described previously 21.” This procedure has a potential risk of circularity. If any of EADB samples were included in the prior studies to genetic those

significant SNPs, the procedure is biased. The testing data would become no longer independent, since they were used to derive those significant SNPs. Any relevant discussion on this topic would be helpful.

We agree with the reviewer that there is a potential for circularity here. Previous data (<https://doi.org/10.1038/s41588-021-00921-z>; [10.1016/S1474-4422\(19\)30435-1](https://doi.org/10.1016/S1474-4422(19)30435-1); <https://doi.org/10.1038/s41588-022-01024-z>) were used to inform which additional imputed SNPs were included in the current study. We elected to include these to ensure the most associated variants in a region were present but would not bias results, as LD pruning thresholds are weak ($r^2=0.75$), meaning another variant in moderate-to-high LD would be included if these variants were excluded. However, to assess whether this inclusion upwardly biased AUCs or SNP associations, we re-ran the best performing ML models (GBMs and NNs) without the additional imputed variants (see p.8, lines 263-265), apart from rs429358, the *APOE*-e4 associated SNP for which association has been well-replicated across many independent datasets. We find that AUCs are not different with the imputed variants removed, highlighting that the inclusion of these does not upwardly bias results (results, page 13, lines 387-389).

- “To give the same information to both PRS and all machine learning models, PRS were derived using summary statistics which were generated in the training set by running a genome-wide association study (GWAS), adjusting for the same confounders in PLINK 320 v2.00a3.3LM 37 (Supplementary section 1.5)”. It is unclear what p-value threshold was used to derive the PRS. Different thresholds would lead to different prediction performances. To perform PRS we utilised LDAK-Bolt-Predict (<https://www.nature.com/articles/s41467-021-24485-y>), which does not perform a traditional pruning and thresholding-based PRS

with a p -value threshold. Instead, it reweights variants using Gaussian priors informed by heritability models, which incorporate minor allele frequency, linkage disequilibrium, and functional annotations, shrinking effect sizes without the need for p -value thresholding. We now highlight this in the methods section (page 10, lines 324-327).

- “The top SNPs identified by machine learning were selected by the permutation-based p -value threshold defined above for MB-MDR ($p_{adj} < 1$), empirically by taking the extreme tail of the distribution for SHAP values ($\mu|SHAP| > 0.0005$) in gradient boosting (Supplementary Figure 3), summed weights ($p_{adj} < 0.05$) for neural networks (Supplementary Figure 5), and by applying the Boruta algorithm 38 (gradient boosting models only). Details on predictor selection are given in Supplementary sections 1.6 and 2.2-2.4.” Different methods for determine relevant features were used among three ML models. It is unclear whether these SNP identification methods are comparable among one another.

We thank the reviewer for raising this point. We acknowledge that different feature selection methods were applied across the three ML models. This choice was driven by the inherent differences in how each model represents feature importance: MB-MDR employs a permutation-based approach suited for detecting SNP-SNP interactions. Gradient Boosting (GBM) relies on SHAP values, which provide an interpretable measure of feature contribution. The Boruta algorithm was additionally used to enhance feature selection robustness. Neural Networks utilize summed weights, a method commonly applied to assess the relative importance of input features in deep learning models.

Despite methodological differences, all approaches aim to identify the most relevant SNPs based on their contribution to the predictive model. To assess their comparability, we

examined the overlap of top genes across methods (page 15, lines 435-440), as approaches may identify different SNPs within the same gene. The methods for SNP selection differ across the machine learning (ML) approaches due to the fundamental differences in how these models operate, making direct comparability of selection methods inherently challenging. This issue is not unique to our study but reflects a broader challenge in the field when comparing outputs from diverse ML frameworks. Importantly, our study is not primarily comparative between ML approaches; the main focus is on identifying genetic signals that replicate in external summary statistics as a unifying validation step, alongside benchmarking predictive performance. We acknowledge these limitations of top-SNP selection methods in the discussion (page 22, lines 597-603).

- “Known lead variants in APOE and BIN1 were the most important predictors in GBM and NNs, with causal SNPs for deriving APOE status identified and ranked as the most important SNPs by both GBMs and NNs (Figure 3).” The meaning of “causal SNPs” is unclear. It would be helpful to provide further clarification.

Here we refer to the SNPs required to derive the e2 and e4 allele status for *APOE*, which are rs7412 and rs429358 respectively. We have corrected the corresponding sentence (page 15, lines 426-429). Here we demonstrate that, unlike GWAS, GBMs and NNs correctly highlight these two e2-e4 SNPs as the most important in the region, and that GBMs derive the correct *APOE* status directly from the SNPs, without requiring the manual coding often used in PRS. The discussion has been updated to clarify this (page 17, lines 475-482).

In summary, the value of the results presented here is unclear, given that they were not replicated yet or their stability was not examined yet. The significance of the work is unclear

either, given that the criteria for generating ML-derived SNPs need further justification. The current work does not support the conclusions and claims. The additional evidence to address the above concerns is needed.

We have now checked both stability and replication, and report putative novel genes which are both stable across repeats and replicate in an external dataset. We have also updated the abstract (page 5, lines 198-205), introduction (page 6, lines 219-230) and discussion (page 15, lines 465-467 and page 23, lines 621-627) to clarify the gap in the literature, how the paper addresses this and how it adds value through identification of novel loci via a complementary approach which maximises discovery from AD genetics, and can be used alongside traditional approaches. We thank the reviewer for their comments, we believe the additional analyses requested by the reviewer have strengthened the paper.

The value of the results presented here is unclear, given that they were not replicated yet or their stability was not examined yet.

We kindly refer to the answer above for stability, replication and added value of results.

The significance of the work is unclear either, given that the criteria for generating ML-derived SNPs need further justification.

We acknowledge that ML-derived SNPs, like novel SNPs identified through GWAS, require further validation and justification. As noted before, comparison of predictive performance and predictor selection across methods is an inherent challenge to the field, and not specific to our paper. However, all variants are now subject to requirements for stability and replication (see above). We believe that these modifications make the significance of the results clear.

Reviewer #1 (Remarks on code availability):

The github repositories were well-documented. I did not test the code.

We thank the reviewer for their comment.

Reviewer #2 (Remarks to the Author):

Cool paper + big and solid team.

We thank the reviewer for their comments and response to each in-kind below.

Some comments:

- "co-dominant" in reality should be dosage or additive.

We have updated this section to clarify that the effects are traditionally additively modelled (page 6, lines 211-214).

- Gridsearch run details or bayes opt params for hyper tune?

Random search was applied for GBMs as they have multiple hyperparameters to tune (Bengio et al., Random Search for Hyper-Parameter Optimization, JMLR, 2012), with grid search used for other approaches. We have clarified this in the methods (page 9, lines 282-284).

- "multiplicative relationship between SNPs" needs more detail regarding allele direction, 0 count compensation (+ 1) and if this is main (adjusting for SNP effects themselves separately) or joint effects. Additionally, some of the algorithms used will be including interactions or tree based proxies with information content representing interactions.

We have now included an example equation and SNP coding in the methods to clarify that models are run as $\text{logit}(y) = \beta_0 + \beta_1\text{SNP}_1 + \beta_2\text{SNP}_2 + \beta_3\text{SNP}_1*\text{SNP}_2$, where SNPs are coded additively (0, 1, 2) and covariates are also included in the model. We agree with the reviewer that all the ML algorithms include some explicit or implicit modelling of interactions and now note this in the manuscript alongside the expanded methods (page 12, lines 367-371).

- "IDUA and DGKQ" also other diseases of the brain like Parkinson's, IIRC.

We thank the reviewer for bringing this to our attention. We have now included references to highlight the IDUA/DGKQ association with multiple brain disorders, including Parkinson's disease (page 19, lines 528-531).

- Please provide more detailed performance stats such as balanced accuracy, sensitivity, specificity as well as prevalence adjusted TPR + NPR etc.

We now include additional classification metrics in Figure 2h. We include accuracy over balanced accuracy as the case/control ratio is evenly split. As predicted values are again adjusted for covariates, we use percentiles to threshold predictions into classes for classification metrics. We apply a standard 50th percentile threshold. We also note that classification metrics are dependent on the threshold applied to quantitative predictions, which is somewhat arbitrary, and hence we still focus on AUCs in the manuscript as they are independent of the threshold.

- Why those algorithms for testing?

We selected these three methods for their prominence in the field, strong performance, and complementary strengths. GBMs are widely recognized for their robust performance across diverse datasets and are particularly effective at optimising metrics such as AUC. Neural networks offer flexibility and allow for the integration of prior knowledge into their architecture, as applied in this study. Meanwhile, MB-MDR is specifically designed for identifying SNP-SNP interactions (<https://doi.org/10.1093/bib/bbr012>). This is now clarified in the discussion (page 16, lines 467-472).

- More details are needed on accounting for covariates. Some people go as far as adjusting genotypes for the covariates then running the training after normalization. How was numeric space normalized for covariates in models, Z scale?

The approach described by the reviewer is exactly what was implemented with gradient boosting, where covariates were regressed-off from the genotypes before training, while neural networks include the covariates in the final layer of the model (with predictions and importance scores taken from non-covariate nodes only). As noted by the reviewer, z-scaling was used to normalise the covariates. This has now been clarified in the methods subsection on “machine learning” (page 8, lines 275-282), with details on the NN architecture given in Supplementary Figure 3.

- Static 70:30 for all algorithms, same samples in each algo test?

Correct, the same samples are present in the train and test splits for each algorithm to ensure performance estimates are comparable across models.

Reviewer #3 (Remarks to the Author):

This is an interesting and informative study in terms of topic and comparing machine learning and statistical methods for analysing genomic data.

The pre-processing and machine learning pipelines are clearly described. The supplementary material also provides more detailed information about the correlation analysis and performance of the models.

The scale of the data used in the study offers statistical power, allowing for reliable assessment of potential confounders. The quality control methods are robust. The use of biologically informed neural network is a good example of how machine learning models can mimic existing information structures. It is also especially interesting to read about the novel SNPs, genes, and pathways that the ML models identified.

The findings are evaluated based on the identified loci and SNPs via the models, and these findings are compared with known literature.

While the results and the proposed models are interesting, it would be helpful if the authors could further clarify how these findings add to the understanding of AD genomics and the current literature in this area.

We thank the reviewer for their detailed review and comments. We respond to each comment individually below.

Regarding the method and results:

- The results of the three machine learning models are compared using a statistical test, and it is reported that they all performed similarly on the test data (AUC=0.69). This statistical test provides interesting insight into the fact that the models predicted the outcomes in the

same direction. However, it would be very informative if the authors could report the prediction task results using specificity sensitivity metrics. This would have helped to better understand the performance of the models.

We now include additional classification metrics in Figure 2h, including accuracy, F1 score, sensitivity and specificity. It should be noted, however, that metrics like sensitivity and specificity depend on the chosen threshold. For this reason, we focus on AUCs in the rest of the paper, as they provide a more threshold-independent evaluation of model performance.

- In relation to the above, the AUC is not high, and especially without including the APOE region, the AUC is around 0.59; the confidence in findings and feature importance extracted from the model to derive the statical findings should also be treated with caution. It would have been useful if further improvements on the models could be made to improve the prediction of the model, and then the feature analysis would have been performed on the best performing model. Showing that the three models (with similar predictive power and similar input) perform similarly does not improve the confidence in the results.

We acknowledge the reviewer's comment regarding moderate AUC values falling short of clinically actionable thresholds (commonly cited as around 0.8). We note that all predictions in genomics are bounded by the heritability of the disease (Wray et al., 2010 - <https://doi.org/10.1371/journal.pgen.1000864>). While we agree that further model improvements could enhance predictive performance, we believe that the greatest improvements would come from augmentations to the data itself (through sequencing data, including of rare variants or inclusion of additional "omics" data), but that this is beyond the scope of the paper. Instead, we focus on maximising insights from existing and readily-available genotyping data. In addition to the robustness and biological relevance of our

findings, we highlight hits that replicate in external datasets (Table 1), that AUCs are stable across random splits (page 13, lines 384-391), and that, in addition to novel loci (page 14, lines 400-406, and Table 1), ML models identify a comprehensive set of well-established AD risk loci (page 13, lines 396-400, and Table 1), which collectively increase confidence and underscore the validity of the findings.

- The presented AUC does not report confidence intervals or variations across different runs. While a 70/30 split is used for training/test and 4-fold cross-validation is applied for hyperparameter tuning, a bootstrapping method could be applied to the training set to demonstrate variations of the metrics across multiple runs.

We thank the reviewer for their comment. We now include confidence intervals for AUCs (page 13, lines 384-387, and annotated as whiskers in Figure 2a). For this we use the confidence interval function in the R pROC package (doi.org/10.1186/1471-2105-12-77) which uses a bootstrapping approach in the test set. Further, we also demonstrate that performance is stable across 4 different random splits (page 13, lines 389-391, and Supplementary Table 4), though computational burden in training models is too large to obtain CIs through a full bootstrapping procedure wherein models are retrained in many resamples.

- In the experiment excluding APOE, it would be useful if the authors had identified APOE carriers of the participants with known AD status and then ran an experiment to show how the prediction results could vary by excluding the APOE SNPs for APOE carrier/non-carriers. We agree with the reviewer and considered running these analyses as part of the original manuscript. However, we believe that fully addressing this question requires a separate

study design where analyses are run separately in all *APOE* strata, especially e44, e4 and e33 carriers separately. The consortium (EADB), which has kindly provided us these data, has a separate e4-stratification analysis group, which is looking at the *APOE*-stratified analyses.

- The authors have indicated that the previous studies using ML had the risk of bias. It would have been helpful if the authors had conducted a post-model prediction analysis to report on the differences in predictions (if any) according to sex, age, and other demographic data.

We have now performed this analysis and included it as part of Figure 2i in the main text.

Demographics are similar between predicted cases and controls across top performing models. Though predicted classes do vary by sex for some models, we note that the absolute difference in age is relatively small, and that sex differences in AD prevalence may be a genuine aspect of the disease.

- The authors mention that (line 605) "Finally, selection of genes for interpretation of models' results was somewhat arbitrary, as there are no standard thresholds for important feature selection in different ML approaches; this may partially explain the incomplete overlap between top SNPs identified between different ML approaches." Would it be possible to choose for example top n features based on the SHAP values and then perform a survey analysis to say out of the "n" top features how many of them were previously known and reported in literature?

We thank the reviewer for their comment and agree that reporting the overlap between top features from models and previously known hits is important. In response to comments from another reviewer on stability and replication, we now highlight model results collectively which are stable across random repeats of a given model (see methods page 11,

lines 338-349, and Table 1). We now specifically check what proportion of genes annotated to these SNPs are previously known and reported in the literature. We report that 22% (19 of 86) of known reported loci are identified here by ML (page 15, lines 429-440). We have also expanded this final point in the discussion to note that a standard framework for extracting top hits across diverse models is a broader challenge in the field, but that the results here do have a common bottleneck in stability across runs and whether their association is seen in other datasets (replication), which we now make a focus of the manuscript (page 22, lines 597-603).

- I'm curious why you think different ML models identified SNPs that were enriched in different cell types. You wrote: "SNPs prioritised by GBM were significantly enriched in microglial regions and showed greater enrichment in astrocytes when modelled without the APOE region. Findings from NN and MBMDR were enriched in synaptic regions." Please explain why you think this is in more detail.

We ran cell type enrichment analysis to investigate whether novel loci are relevant to disease. We initially ran these separately for each ML approach to confirm our observations from an earlier study ([10.1016/j.schres.2022.06.006](https://doi.org/10.1016/j.schres.2022.06.006)) that different patterns are to be expected with different ML approaches. Though we still hold that this is interesting, we have taken methods together in a focus on stability of models and replicability of association, as requested by another reviewer, and to simplify the message now report only enrichment on the combined reported SNPs.

- In your Quality Control section, you write: "For ML based analysis, we limited the data to genotyped (non-imputed) variants. We additionally integrated imputed lead independent

SNPs previously reported as genome-wide significant after converting to best guess genotypes” – these sentences seem contradictory; did you use imputed values or not?

The genotype data had been imputed as part of Bellenguez et al.

(<https://doi.org/10.1038/s41588-022-01024-z>) and is available upon request in both genotyped and imputed formats. We primarily use directly genotype data (unimputed) to ensure high quality well-typed variants. However, this runs the risk that established AD-associated SNPs may not be present in the data. We therefore extracted imputed values for established loci where they were absent in the genotyped data, and merged these into a final dataset that is still primarily genotyped data. We have clarified this in the methods section (page 7, lines 251-265). We also now refer to Supplementary Figure 1 which gives a clear flow diagram for SNP selection procedures.

- Please ensure all abbreviations are appropriately defined. Ex) MAF is not listed in parentheses after “minor allele frequency”.

We have corrected abbreviations through the manuscript.

- The authors have assessed the power of ML by measuring the correlation between ML and traditional models – does this necessarily measure the quality of ML as a novel approach to modelling disease risk, or simply how similar it is to existing models? As far as I understand, the purpose of this work was to determine how ML can surpass traditional methods in modelling disease, couldn't that come at a cost of its correlation with existing methods?

We wholeheartedly agree with the reviewer that correlation with existing models alone does not demonstrate the quality of ML as a novel approach. Here we use this as an important benchmark, as PRS approaches are widespread and frequently applied, but also

focus on handling of the crucial *APOE* region and identification of known and novel loci to evaluate models. We also agree completely with the reviewer that identifying novel associations with a disease and modelling aspects which have previously been ignored will come at a cost of correlation with standard linear models like PRS. We have updated the discussion to reflect this (page 20, lines 548-552). We have also updated the abstract (page 5, lines 198-205), introduction (page 6, lines 219-230) and discussion (page 15, lines 465-467 and page 23, lines 621-627) to clarify the gap in the literature, how the paper addresses this and how it adds value through identification of novel loci via a complementary approach which maximises discovery from AD genetics, and can be used alongside traditional approaches.

- The use of federated learning models is mentioned as one of the future directions. The privacy requirement is clear, but in cases where the data can be combined (such as this example), there are other ways to improve the homogeneity of the data across multiple sites/sources. It would have been helpful if some of these approaches would have been mentioned.

We thank the reviewer for this suggestion. We have expand the relevant section to include approaches to data harmonisation in centralised learning paradigms (page 22, lines 603-619).

Reviewer #3 (Remarks on code availability):

I did not review the code, but the GitHub repositories are well-structured and the DAXOS: Dask and XGBoost on SNPs includes the installation, usage and test instructions. This will

help with the reproducibility of the work.

Reviewer #4 (Remarks to the Author):

I co-reviewed this manuscript with one of the reviewers who provided the listed reports.

This is part of the Nature Communications initiative to facilitate training in peer review and to provide appropriate recognition for Early Career Researchers who co-review manuscripts.

Reviewer #4 (Remarks on code availability):

N/A

We thank the reviewer for taking the time to co-review the manuscript.

Response to reviewers

Reviewer #1 (Remarks to the Author)

My comments have been properly addressed.(Remarks on code availability)

We thank the reviewer for their helpful comments in revising the manuscript.

Reviewer #2 (Remarks to the Author)

Thanks for the revisions.

- Still not sure if 0 count compensation on the formal interaction analysis was used when the regression was run. Without this, dynamic range is flattened for all major hmozygotes.

We have updated the manuscript to reflect that a +1 zero count compensation was used in the formal interaction analyses (page 25, line 636-637).

- Please be clear that you are using probabilistic AUCs not including specific cut off values.

We have updated the manuscript to reflect this (page 22, lines 546-549).

- Code and data availability could be a little clearer, but we all understand painful logistics. Good luck!

We thank the reviewer for their helpful comments.

M.A. Nalls(Remarks on code availability)

These are packages but not the minutae of the paper itself. Generally seems fine.

Reviewer #3 (Remarks to the Author)

I would like to thank the authors for revising the paper and for responding to the review comments. The authors have responded to most of our comments and concerns.

Only two minor suggestions:

1. The sensitivity and specificity are relatively low, and the authors also agree that running an experiment to show how the prediction results could vary by excluding the APOE SNPs for APOE carrier/non-carriers would be helpful. The authors' response is added below.

"We agree with the reviewer and considered running these analyses as part of the original manuscript. However, we believe that fully addressing this question requires a separate study design where analyses are run separately in all APOE strata, especially e44, e4 and e33 carriers separately. The consortium (EADB), which has kindly provided us these data, has a separate e4-stratification analysis group, which is looking at the APOE-stratified analyses."

In this case, please add a subsection to the discussion section and clearly state this as a limitation to their current study and a potential area for future work.

We thank the reviewer for their comments, which have helped strengthen the manuscript. We have included a clear statement in the discussion to list this as a limitation and area for future work (page 18, lines 465-469).

2. The differences based on the female/male split are shown in Figure 2 (i); the panel is difficult to see/interpret; it would be good if the authors also included one or two sentences about the results of this new experiment in the paper.

We thank the reviewer for highlighting this. We have now added to the manuscript to aid interpretation of the female/male split (page 9, 249-252).

(Remarks on code availability)

n/a